# ManyDG: Many-domain Generalization for Healthcare Applications

**Chaoqi Yang**[1], **M. Brandon Westover**[2,3], **Jimeng Sun**[1]
[1]University of Illinois Urbana-Champaign
[2]Harvard Medical School
[3]Beth Israel Deaconess Medical Center
`{chaoqiy2}@illinois.edu`

## Abstract

The vast amount of health data has been continuously collected for each patient, providing opportunities to support diverse healthcare predictive tasks such as seizure detection and hospitalization prediction. Existing models are mostly trained on other patients' data and evaluated on new patients. Many of them might suffer from poor generalizability. One key reason can be overfitting due to the unique information related to patient identities and their data collection environments, referred to as *patient covariates* in the paper. These patient covariates usually do not contribute to predicting the targets but are often difficult to remove. As a result, they can bias the model training process and impede generalization.

In healthcare applications, most existing domain generalization methods assume a small number of domains. In this paper, considering the diversity of patient covariates, we propose a new setting by *treating each patient as a separate domain* (leading to many domains). We develop a new domain generalization method `ManyDG`[1], that can scale to such many-domain problems. Our method identifies the patient domain covariates by mutual reconstruction, and removes them via an orthogonal projection step. Extensive experiments show that `ManyDG` can boost the generalization performance on multiple real-world healthcare tasks (e.g., 3.7% Jaccard improvements on MIMIC drug recommendation) and support realistic but challenging settings such as insufficient data and continuous learning.

## 1 Introduction

The remarkable ability of functional approximation (Hornik et al., 1989) can be a double-edged sword for deep neural nets (DNN). Standard empirical risk minimization (ERM) models that minimize average training loss are vulnerable in applications where the model tends to learn *spurious correlations* (Liu et al., 2021; Wiles et al., 2022) between *covariates* (opposed to the *causal factor* (Gulrajani & Lopez-Paz, 2021)) and the targets during training. Thus, these models often fail on new data that do not have the same spurious correlations. In clinical settings, Perone et al. (2019); Koh et al. (2021); Castro et al. (2020) have shown that a prediction model trained on a set of hospitals often does not work well for another hospital due to *covariate shifts*, e.g., different devices, clinical procedures, and patient population.

Unlike previous settings, this paper targets the *patient covariate shift* problem in the healthcare applications. Our motivation is that patient-level data unavoidably contains some unique personalized characteristics (namely, *patient covariates*), for example, different environments (Zhao et al., 2017) and patient identities, which are usually independent of the prediction targets, such as sleep stages (Zhao et al., 2017) and seizure disease types (Hirsch et al., 2021). These patient covariates can cause spurious correlations in learning a DNN model.

For example, in the electroencephalogram (EEG) sleep staging task, the patient EEG recordings may be collected from different environments, e.g., in sleep lab (Terzano et al., 2002) or at home (Kemp et al., 2000). These measurement differences (Zhao et al., 2017) can induce noise and biases. It is also common that the label distribution per patient can be significantly different. A patient with

---

[1]Code is available at https://github.com/ycq091044/ManyDG.

insomnia (Rémi et al., 2019) could have more awake stages than ordinary people, and elders tend to have fewer rapid eye movement (REM) stages than teenagers (Ohayon et al., 2004). Furthermore, these patient covariates can be even more harmful when dealing with insufficient training data.

In healthcare applications, recent domain generalization (DG) models are usually developed in cross-institute settings to remove each hospital's unique covariates (Wang et al., 2020; Zhang et al., 2021; Gao et al., 2021; Reps et al., 2022). Broader domain generalization methods were built with various techniques, including style-based data augmentations (Nam et al., 2021; Kang et al., 2022; Volpi et al., 2018), episodic meta-learning strategies (Li et al., 2018a; Balaji et al., 2018; Li et al., 2019) and domain-invariant feature learning (Shao et al., 2019) by heuristic metrics (Muandet et al., 2013; Ghifary et al., 2016) or adversarial learning (Zhao et al., 2017). Most of them (Chang et al., 2019; Zhou et al., 2021) limit the scope within CNN-based models (Bayasi et al., 2022) and batch normalization architecture (Li et al., 2017) on image classification tasks.

Unlike previous works that assume a small number of domains, this paper considers a new setting for learning generalizable models from many more domains. To our best knowledge, we are the first to propose the *many-domain generalization* problem: In healthcare applications, we handle the diversity of patient covariates by *modeling each patient as a separate domain*. For this new many-domain problem, we develop an end-to-end domain generalization method, which is trained on a pair of samples from the same patient and optimized via a Siamese-type architecture. Our method combines mutual reconstruction and orthogonal projection to explicitly remove the patient covariates for learning (patient-invariant) label representations. We summarize our main contributions below:

- We propose a new **many-domain generalization problem** for healthcare applications by treating **each patient as a domain.** Our setting is challenging: We handle many more domains (e.g., 2,702 in the seizure detection task) compared to previous works (e.g., six domains in Li et al. (2020)).
- We propose a **many-domain generalization method** `ManyDG` for the new setting, motivated by a latent data generative model and a factorized prediction model. Our method explicitly captures the domain and domain-invariant label representation via orthogonal projection.
- We evaluate our method `ManyDG` on **four healthcare datasets and two realistic and common clinical settings**: (i) insufficient labeled data and (ii) continuous learning on newly available data. Our method achieves consistently higher performance against the best baseline (e.g., $3.7\%$ Jaccard improvements in the benchmark MIMIC drug recommendation task).

## 2 RELATED WORKS

**Domain Generalization (DG)**   The main difference between DG (Wang et al., 2022) (also called multi-source domain generalization, MSDG) and domain adaptation (Wilson & Cook, 2020) is that the former cannot access the test data during training and is thus more challenging.

This paper focuses on the DG setting, for which recent methods are mostly developed from image classification. They can be broadly categorized into three clusters (Yao et al., 2022): (i) Style augmentation methods (Nam et al., 2021; Kang et al., 2022) assume that each domain is associated with a particular style, and they mostly manipulate the style of raw data (Volpi et al., 2018; Zhou et al., 2020b) or the statistics of feature representations (the mean and standard deviations) (Li et al., 2017; Zhou et al., 2021) and enforces to predict the same class label; (ii) Domain-invariant feature learning (Li et al., 2018b; Shao et al., 2019) aims to remove the domain information from the high-level features by heuristic objectives (such as MMD metric (Muandet et al., 2013; Ghifary et al., 2016), Wasserstein distance (Zhou et al., 2020a)), conditional adversarial learning (Zhao et al., 2017; Li et al., 2018c) and contrastive learning (Yao et al., 2022); (iii) Meta learning methods (Li et al., 2018a) simulate the train/test domain shift during the episodic training (Li et al., 2019) and optimize a meta objective (Balaji et al., 2018) for generalization. Zhou et al. (2020a) used an ensemble of experts for this task. For a more holistic view, two recent works have provided systematic evaluations (Gulrajani & Lopez-Paz, 2021) and analysis (Wiles et al., 2022) on existing algorithms.

Our paper considers a new setting by modeling each patient as a separate domain. Thus, we deal with many more domains compared to all previous works (Peng et al., 2019). This challenging setting also motivates us to capture the domain information explicitly in our proposed method.

**Domain Generalization in Healthcare Applications**   Many healthcare tasks aim to predict some clinical targets of interest for one patient: During each hospital visit, such as estimating the risk of developing Parkinson's disease (Makarious et al., 2022) and sepsis (Gao et al., 2021), predicting the

endpoint of heart failure (Chu et al., 2020), recommending the prescriptions based on the diagnoses (Yang et al., 2021a;b; Shang et al., 2019); During each measurement window, such as identifying the sleep stage from a 30-second EEG signal (Biswal et al., 2018; Yang et al., 2021c), detecting the IIIC seizure patterns Jing et al. (2018); Ge et al. (2021). In such tasks, *every patient can generate multiple data samples*, and each data unavoidably contains patient-unique covariates, which may bias the training performance. Most existing works (Wang et al., 2020; Zhang et al., 2021; Gao et al., 2021) consider model generalization or adaptation across multiple clinical institutes (Reps et al., 2022; Zhang et al., 2021) or different time frames (Guo et al., 2022) with heuristic assumptions such as linear dependency (Li et al., 2020) in the feature space. Fewer works attempt to address the patient covariate shift. For example, Zhao et al. (2017) trained a conditional adversarial architecture with multi-stage optimization to mitigate environment noise for radio frequency (RF) signals. However, their approach requires a large number of parameters (linear to the number of domains). In our setting with more patient domains, their method can be practically less effective, as we will show in the experiments. Compared to Zhao et al. (2017), our method can handle many more domains with a fixed number of learnable parameters and follows end-to-end training procedures.

## 3 MANY-DOMAIN GENERALIZATION FOR HEALTHCARE APPLICATIONS

The section is organized by: (i) We first formulate the main-domain generalization problem in Section 3.1; (ii) Then, we motivate our model design by assuming the data generation process in Section 3.2; (iii) In Section 3.3, we simultaneously develop model architecture and objectives as they are interdependent; (iv) We summarize the training and inference procedure in Section 3.4.

### 3.1 PROBLEM DEFINITION: MANY-DOMAIN GENERALIZATION

We denote $\mathcal{D}$ as the *domain* set of training (e.g., a set of training patients, each as a domain). In the setting, we assume the size of $\mathcal{D}$ is large, e.g., $|\mathcal{D}| > 50$. Each domain $d \in \mathcal{D}$ has multiple data (e.g., one patient can generate many data samples), and thus we use $\mathcal{S}^d$ to denote the *sample* set of domain $d$. The notation $\mathbf{x}_i^d$ represents the $i$-th sample of domain $d$ (e.g., a 30-second EEG sleep signal), which is associated to label $y_i^d$ (e.g., the "rapid eye movement" sleep stage). The many-domain generalization problem asks to find a mapping, $\mathbf{f} : \mathbf{x} \rightarrow y$, based on the training set $\{(\mathbf{x}_i^d, y_i^d) : i \in \mathcal{S}^d, d \in \mathcal{D}\}$, such that given new domains $\mathcal{D}'$ (e.g., a new set of patients), we can infer the labels of $\{\mathbf{x}_i^{d'} : i \in \mathcal{S}^{d'}, d' \in \mathcal{D}'\}$. We provide a notation table in Appendix A.1.

### 3.2 MOTIVATIONS ON METHOD ARCHITECTURE DESIGN

This paper targets the *many-domain generalization problem* in healthcare applications by treating each patient as an individual domain. The problem is challenging since we deal with many more domains than previous works. Our setting is also unique: All the domains are induced by individual patients, and thus they naturally follow a meta domain distribution formulated below.

**Data Generative Model**  We assume the training data is generated from a two-stage process. First, each domain is modeled as a latent factor $\mathbf{z}$ sampled from some *meta domain distribution* $p(\cdot)$; Second, each data sample is from a *sample distribution* conditioned on the domain $\mathbf{z}$ and class $y$:

$$\mathbf{z} \sim p(\cdot), \quad \mathbf{x} \sim p(\cdot|\mathbf{z}, y). \tag{1}$$

**Factorized Prediction Model**  Given the generated sample $\mathbf{x}$, we want to uncover its true label using the posterior $p(y|\mathbf{x})$. The quantity can be factorized by the domain factor $\mathbf{z}$ as

$$p(y|\mathbf{x}) = \int p(y, \mathbf{z}|\mathbf{x})d\mathbf{z} = \int p(y|\mathbf{x}, \mathbf{z})p(\mathbf{z}|\mathbf{x})d\mathbf{z}. \tag{2}$$

**Motivations on Method Architecture Design**  Thus, a straightforward solution for addressing the many-domain generalization problem is: Given a data sample $\mathbf{x}$ in the training set, we first identify the latent domain factor $\mathbf{z}$ and then build a *factorized prediction model* to uncover the true label $y$ in consideration of the underlying *data generative model*.

- **Motivation 1:** By the form of the prediction model, we need (i) a *domain encoder* $p(\mathbf{z}|\mathbf{x})$; (ii) a *label predictor* $p(y|\mathbf{x}, \mathbf{z})$. Cascading the encoder and predictor can generate the posterior label $y$, which motivates the forward architecture of our method.

- **Motivation 2:** To guide the learning process of $\mathbf{z}$ (i.e., ensuring it is domain representation), we consider reconstruction (inspired by VAE (Kingma & Welling, 2013)). By the form of the data generative model $p(\mathbf{x}'|\mathbf{z}, y')$, we use $\mathbf{z}$ and another sample $(\mathbf{x}', y')$ to calculate reconstruction loss.

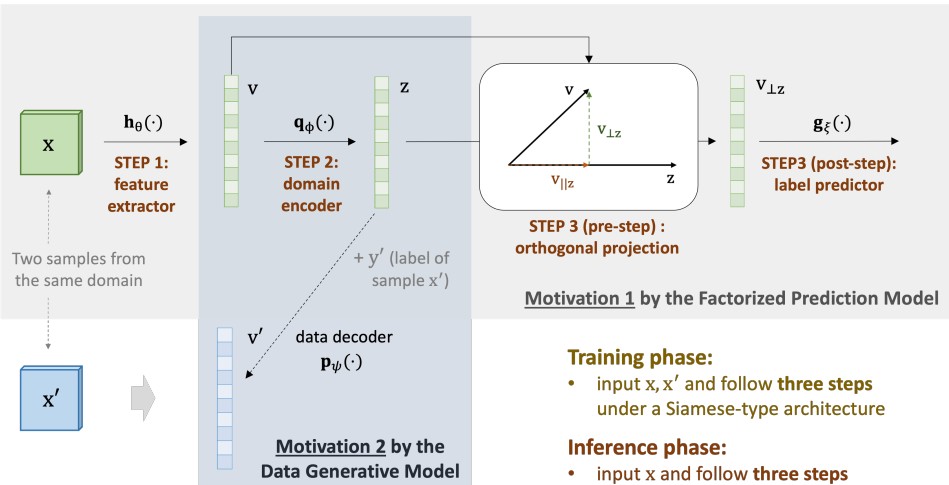

Figure 1: `ManyDG` Framework. The forward architecture (solid lines) for both training and inference follows **three steps** (inspired by the factorized prediction model): feature extraction, domain encoding, and label prediction via orthogonal projection. During training, we input two samples from the same domain, a unique design in our method is that we use the estimated domain $\mathbf{z}$ (from $\mathbf{x}$) and the true label $y'$ (from $\mathbf{x}'$) to reconstruct the encoded feature $\mathbf{v}'$ (from $\mathbf{x}'$). Conversely, we reconstruct $\mathbf{v}$ by $\mathbf{z}'$ and $\mathbf{y}$. This **mutual reconstruction** design enables to learn better domain factors $\mathbf{z}, \mathbf{z}'$. The feature $\mathbf{v}$ encode domain and label information along approximately different embedding dimensions, which can be decomposed into $\mathbf{v}_{||\mathbf{z}}$ and $\mathbf{v}_{\perp\mathbf{z}}$ respectively via **orthogonal projection**.

### 3.3    ManyDG: Method to handle many-domain generalization

We are motivated to develop our method `ManyDG` (shown in Figure 1) in a principled way.

**STEP 1:  Universal Feature Extractor**    Before building the prediction model, it is common to first apply a learnable feature extractor, $\mathbf{h}_\theta(\cdot) : \mathbf{x} \mapsto \mathbf{v}$, with parameter $\theta$, on the input sample $\mathbf{x}$. The output feature representation is $\mathbf{v}$,

$$\mathbf{v} = \mathbf{h}_\theta(\mathbf{x}). \tag{3}$$

The encoded feature $\mathbf{v}$ can include both the domain information (i.e., patient covariates in our applications) and the label information. Also, $\mathbf{h}_\theta(\cdot)$ could be any neural architecture, such as RNN (Rumelhart et al., 1985; Hochreiter & Schmidhuber, 1997; Chung et al., 2014), CNN (LeCun et al., 1995; Krizhevsky et al., 2017), GNN (Gilmer et al., 2017; Kipf & Welling, 2016) or Transformer (Vaswani et al., 2017). In our experiments, we specify the choices of feature encoder networks based on the healthcare applications.

**STEP 2: Domain Encoder**    Following **Motivation 1**, we parameterize the domain encoder $p(\mathbf{z}|\mathbf{x})$ by an *encoder network* $\mathbf{q}_\phi(\cdot)$ with parameter $\phi$. Since $\mathbf{h}_\theta(\cdot)$ already encode the feature embedding $\mathbf{v}$, we apply $\mathbf{q}_\phi(\cdot)$ on $\mathbf{v}$ (not the raw input $\mathbf{x}$) to estimate the domain factor,

$$\mathbf{z} = \mathbf{q}_\phi(\mathbf{v}). \tag{4}$$

Note that though Equation (2) has a probabilistic form, we use a deterministic three-layer neural network $\mathbf{q}_\phi(\cdot)$ for the prediction task. We ensure that $\mathbf{v}$ and $\mathbf{z}$ have the same dimensionality.

**Objective 1: Mutual Reconstructions**    To ensure that $\mathbf{z}$ represents the true latent domain of $\mathbf{x}$, we postpone the forward architecture design. This section proposes a mutual reconstruction task as our first objective to guide the learning process of $\mathbf{z}$, following **Motivation 2**:

- For the current sample $(\mathbf{x}, y)$ with feature $\mathbf{v}$ and domain factor $\mathbf{z}$, we take **a new sample** $(\mathbf{x}', y')$ **from the same domain/patient in the training set**.

- We obtain the feature and the domain factor of this new sample, denoted by $\mathbf{v}'$, $\mathbf{z}'$:

$$\mathbf{v}' = \mathbf{h}_\theta(\mathbf{x}'), \quad \mathbf{z}' = \mathbf{q}_\phi(\mathbf{v}'). \tag{5}$$

- Since $\mathbf{x}$ and $\mathbf{x}'$ share the same domain, we want to use the domain factor of $\mathbf{x}$ and the label of $\mathbf{x}'$ to reconstruct the feature of $\mathbf{x}'$. We utilize the decoder $p(\mathbf{x}|\mathbf{z}, y)$, parameterized by $\mathbf{p}_\psi(\cdot)$.

$$\hat{\mathbf{v}}' = \mathbf{p}_\psi(\mathbf{z}, y'). \tag{6}$$

Note that, we consider reconstructing the feature vector $\mathbf{v}'$ not the raw input $\mathbf{x}'$. Here, the decoder $\mathbf{p}_\psi(\cdot)$ is also a non-linear projection. We implement it as a three-layer neural network in the experiments. Conversely, we reconstruct $\mathbf{v}$ by the domain factor of $\mathbf{x}'$ and the label of $\mathbf{x}$,

$$\hat{\mathbf{v}} = \mathbf{p}_\psi(\mathbf{z}', y). \tag{7}$$

- The reconstruction loss is formulated as mean square error (MSE), commonly used for raw data reconstruction (Kingma & Welling, 2013). In our case, we reconstruct parameterized vectors and further follow (Grill et al., 2020) to use the $L_2$-normalized MSE, equivalent to the cosine distance,

$$\mathcal{L}_{\text{rec}} = -\langle \frac{\mathbf{v}}{\|\mathbf{v}\|}, \frac{\hat{\mathbf{v}}}{\|\hat{\mathbf{v}}\|} \rangle - \langle \frac{\mathbf{v}'}{\|\mathbf{v}'\|}, \frac{\hat{\mathbf{v}}'}{\|\hat{\mathbf{v}}'\|} \rangle. \tag{8}$$

Here, $\langle \cdot, \cdot \rangle$ is the notation for vector inner product and $\| \cdot \|$ is for $L_2$-norm.

**Remark for Mutual Reconstruction** We justify that this objective can guide $\mathbf{z}$ to be the true domain factor of $\mathbf{x}$. Follow Equation (6): First, the reconstructed $\hat{\mathbf{v}}'$ should contain the domain information (which $y'$ as input cannot provide), and thus $\mathbf{z}$ **as another input will strive to preserve the domain information** from $\mathbf{v}$ (in Equation (4)) for the reconstruction; Second, if two samples have different labels (i.e., $y \neq y'$), then $\mathbf{z}$ **will not preserve label information** from $\mathbf{v}$ since it is not useful for the reconstruction; Third, even if two samples have the same labels (i.e., $y = y'$), the input $y'$ in Equation (6) will provide more direct and all necessary label information for reconstruction and thus **discourages $\mathbf{z}$ to inherit any label-related information** from $\mathbf{v}$. Thus, $\mathbf{z}$ is conceptually the true domain representation. Empirical evaluation verifies that $\mathbf{z}$ satisfies the properties of being domain representation, as we show in Section 4.4 and Appendix B.4.

**Objective 2: Similarity of Domain Factors** Further, to ensure the consistency of the learned latent domain factors, we enforce the similarity of $\mathbf{z}$ and $\mathbf{z}'$ (from $\mathbf{x}, \mathbf{x}'$ respectively), since they refer to the same domain. Here, we also use the cosine distance to measure the similarity of two latent factors,

$$\mathcal{L}_{\text{sim}} = -\langle \frac{\mathbf{z}}{\|\mathbf{z}\|}, \frac{\mathbf{z}'}{\|\mathbf{z}'\|} \rangle. \tag{9}$$

After designing these objectives, we go back and follow **Motivation 1** again to finish the architecture design. So far, for the given sample $\mathbf{x}$, we have obtained the universal feature vector $\mathbf{v}$ *that contains both the domain and label semantics*, and the domain factor $\mathbf{z}$ *that only encodes the domain information.* Previous work (Bousmalis et al., 2016) suggests that explicitly modeling domain information can improve model's ability for learning domain-invariant features (i.e., label information). Inspired by this, *the designing goal of our label predictor $p(y|\mathbf{x}, \mathbf{z})$ is to extract the domain-invariant label information from $\mathbf{v}$ with the help of $\mathbf{z}$, for uncovering the true label of $\mathbf{x}$.* For the label predictor $p(y|\mathbf{x}, \mathbf{z})$, we also start from the feature space $\mathbf{v}$ (not raw input $\mathbf{x}$) and parameterize it as a *predictor network*, $\mathbf{g}_\xi(\cdot) : \mathbf{v}, \mathbf{z} \mapsto y$, with parameter $\xi$. To achieve this goal, we rely on the premise that $\mathbf{v}$ *and* $\mathbf{z}$ *are in the same embedding space*, which requires **Objective 3** below.

**Objective 3: Maximum Mean Discrepancy** The Maximum Mean Discrepancy (MMD) loss is commonly used to reduce the distribution gap (Muandet et al., 2013; Ghifary et al., 2016; Li et al., 2018b; Kang et al., 2022). In our model, we use it for aligning the embedding space of $\mathbf{v}$ and $\mathbf{z}$. We consider a normalized version by using the feature norm to re-scale the magnitude in optimization,

$$\mathcal{L}_{\text{MMD}} = \frac{\text{MMD}^2(\mathbf{z}_\mu, \mathbf{v}_\mu)}{\|\mathsf{sg}(\mathbf{v}_\mu)\|_{\mathcal{F}}^2} = \frac{\|\mathbf{z}_\mu - \mathbf{v}_\mu\|_{\mathcal{F}}^2}{\|\mathsf{sg}(\mathbf{v}_\mu)\|_{\mathcal{F}}^2}. \tag{10}$$

Here, $\mathsf{sg}(\cdot)$ means to stop gradient back-propagation for this quantity. We use the norm $\mathcal{F}$ induced by the inner product. The mean values are calculated over the data batch $\mathbf{z}_\mu = \mathbb{E}[\mathbf{z}]$, $\mathbf{v}_\mu = \mathbb{E}[\mathbf{v}]$.

**STEP 3: Label Predictor via Orthogonal Projection**    To this end, we can proceed to design the label predictor network $\mathbf{g}_\xi(\cdot)$ and complete the forward architecture. To obtain the robust transferable features that are invariant to the domains, Bousmalis et al. (2016) uses soft constraints to learn orthogonal domain and label subspaces in domain adaptation settings. A recent paper (Shen et al., 2022) further shows that the universal feature representation learned from contrastive pre-training contains domain and label information along approximately different dimensions.

Inspired by this, we consider the *orthogonal projection* as a pre-step of the predictor (as shown in Figure 1). We visually illustrate the orthogonal projection step in Figure 2.

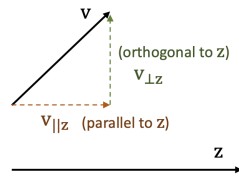

Figure 2: Decomposition

Since $\mathbf{v}$ and $\mathbf{z}$ are in the same space, we decompose $\mathbf{v}$ into two parts (by basic vector arithmetics): A component parallel to $\mathbf{z}$, denoted as $\mathbf{v}_{||\mathbf{z}}$, which will only contain the domain information,

$$\mathbf{v}_{||\mathbf{z}} = \mathbf{z} \cdot \Big\langle \frac{\mathbf{v}}{\|\mathbf{z}\|}, \frac{\mathbf{z}}{\|\mathbf{z}\|} \Big\rangle; \tag{11}$$

Another component orthogonal to $\mathbf{z}$, denoted as $\mathbf{v}_{\perp\mathbf{z}}$, which contains the remaining information, i.e., the label representation,

$$\mathbf{v}_{\perp\mathbf{z}} = \mathbf{v} - \mathbf{v}_{||\mathbf{z}}. \tag{12}$$

This orthogonal projection step is *non-parametric*. The rationale of this step is that we empirically find our universal feature extractor $\mathbf{h}_\theta(\cdot)$ tends to generate $\mathbf{v}$ that contains both domain and label information approximately in different embedding dimensions (shown in Section 4.4). The orthogonal projection step can leverage this property to remove domain covariate information and extract better invariant features $\mathbf{v}_{\perp\mathbf{z}}$ for prediction. Note that the orthogonal projection step is also suitable for use with **Objective 2** since only the angle of vector $\mathbf{z}$ is needed for finding the orthogonal component.

After orthogonal projection, we apply the predictor network $\mathbf{g}_\xi(\cdot)$ (operates on space $\mathbf{v}$) as a post-step to parameterize the label predictor $p(y|\mathbf{x}, \mathbf{z})$, which completes the forward architecture,

$$p(y|\mathbf{x}, \mathbf{z}) = \mathbf{g}_\xi(\mathbf{v}_{\perp\mathbf{z}}). \tag{13}$$

In the experiment, we implement $\mathbf{g}_\xi(\cdot)$ as one fully connected (FC) layer without the bias term, which is essentially a parameter matrix. We also use a temperature hyperparameter $\tau$ (He et al., 2020) to re-scale the output vector before the softmax activation.

**Objective 4: Cross-entropy Loss**    The above objective functions are essentially used to regularize the domain factor $\mathbf{z}$. Finally, we consider the cross-entropy loss, which adds label supervision.

$$\mathcal{L}_{\text{sup}} = -\log p(y|\mathbf{x}, \mathbf{z}). \tag{14}$$

Given that all the objectives are scaled properly, we choose their unweighted linear combination as the final loss to avoid extra hyperparameters. In Appendix B.5, we show that the weighted version can have marginal improvements over our unweighted version.

$$\mathcal{L}_{\text{final}} = \mathcal{L}_{\text{sup}} + \mathcal{L}_{\text{MMD}} + \mathcal{L}_{\text{rec}} + \mathcal{L}_{\text{sim}}. \tag{15}$$

### 3.4    Training and inference pipeline of ManyDG

During training, we input a pair of data $\mathbf{x}, \mathbf{x}'$ from the same domain and follow three steps under a Siamese-type architecture. Four objective functions are computed batch-wise on two sides (implementation of our "double" training loader can refer to Appendix A.2) to optimize parameters $\theta$, $\phi$, $\xi$, and $\psi$. During inference, we directly follow three steps to obtain the predicted labels.

## 4    Experiments

This section presents the following experiments to evaluate our model comprehensively. Data processing files, codes, and instructions are provided in https://github.com/ycq091044/ManyDG.

- **Evaluation on health monitoring data**: We evaluate our ManyDG on multi-channel EEG seizure detection and sleep staging tasks. Both are formulated as multi-class classifications.

- **Evaluation on EHR data**: We conduct visit-level drug recommendation and readmission prediction on EHRs. The former is multi-label classification, and the latter is binary classification.

- **Verifying the orthogonality property**: We quantitatively show that the encoded features from $\mathbf{h}_\theta(\cdot)$ indeed contain domain and label information approximately along different dimensions.

- **Two realistic healthcare scenarios**: We further show the benefits of our `ManyDG` when (i) training with insufficient labeled data and (ii) continuous learning on newly collected patient data.

## 4.1 EXPERIMENTAL SETUP

**Baselines**   We compare our model against recent baselines that are designed from different perspectives. All models use the same backbone network as the feature extractor. We add a non-linear prediction head after the backbone to be the *(Base)* model. Conditional adversarial net *(CondAdv)* (Zhao et al., 2017) concatenates the label probability and the feature embedding to predict domains in an adversarial way. Domain-adversarial neural networks *(DANN)* (Ganin et al., 2016) use gradient reversal layer for domain adaptation, and we adopt it for the generalization setting by letting the discriminator only predict training domains. Invariant risk minimization *(IRM)* (Arjovsky et al., 2019) learns domain invariant features by regularizing squared gradient norm. Style-agnostic network *(SagNet)* (Nam et al., 2021) handles domain shift by perturbing the mean and standard deviation statistics of the feature representations. Proxy-based contrastive learning *(PCL)* (Yao et al., 2022) build a new supervised contrastive loss from class proxies and negative samples. Meta-learning for domain generalization *(MLDG)* (Li et al., 2018a) adopts the model-agnostic meta learning (MAML) (Finn et al., 2017) framework for domain generalization.

**Environments and Implementations**   The experiments are implemented by Python 3.9.5, Torch 1.10.2 on a Linux server with 512 GB memory, 64-core CPUs (2.90 GHz, 512 KB cache size each), and two A100 GPUs (40 GB memory each). For each set of experiments in the following, we split the dataset into train / validation / test by ratio $80\%{:}10\%{:}10\%$ under five random seeds and re-run the training for calculating the mean and standard deviation values. More details, including backbone architectures, hyperparameters selection and datasets, are clarified in Appendix A.4.

## 4.2 RESULTS ON HEALTH MONITORING DATA: SEIZURE DETECTION AND SLEEP STAGING

**Healthcare Monitoring Datasets**   Electroencephalograms (EEGs) are multi-channel monitoring signals, collected by placing measurable electronic devices on different spots of brain surfaces. *Seizure* dataset is from Jing et al. (2018); Ge et al. (2021) for detecting ictal-interictal-injury-continuum (IIIC) seizure patterns from electronic recordings. This dataset contains 2,702 patient recordings collected either from lab setting or at patients' home, and each recording can be split into multiple samples, which leads to 165,309 samples in total. Each sample is a 16-channel 10-second signal, labeled as one of the six seizure disease types: ESZ, LPD, GPD, LRDA, GRDA, and OTHER, by well-trained physicians. *Sleep-EDF Cassette* (Kemp et al., 2000) has in total 78 subject overnight recordings and can be further split into 415,089 samples of 30-second length. The recordings are sampled at 100 Hz at home, and we extract the Fpz-Cz/Pz-Oz channels as the raw inputs to the model. Each sample is classified into one of the five stages: awake (Wake), rapid eye movement (REM), and three sleep states (N1, N2, N3). The labels are downloaded along with the dataset.

**Backbone and Metrics**   We use short-time Fourier transforms (STFT) for both tasks to obtain the spectrogram and then cascade convolutional blocks as the backbone, adjusted from Biswal et al. (2018); Yang et al. (2021c). Model architecture is reported in Appendix A.4. We use *Accuracy*, Cohen's *Kappa* score and *Avg. F1* score over all classes (equal weights for each class) as the metrics.

**Results**   The results are reported in Table 1 (we also report the running time in Appendix B.2). We can conclude that on the seizure detection task, our method gives around $1.8\%$ relative improvements on Kappa score against the best baseline and mild improvements on sleep staging. SagNet and PCL are two very recent end-to-end baselines that relatively perform better among all the baselines. The Base model also gives decent performance, which agrees with the findings in Gulrajani & Lopez-Paz (2021) that vanilla ERM models may naturally have strong domain generalization ability.

Table 1: Result comparison on health monitoring datasets

| Models | Seizure Detection | | | Sleep Staging | | |
|---|---|---|---|---|---|---|
| | Accuracy | Kappa | Avg. F1 | Accuracy | Kappa | Avg. F1 |
| **Base** | $0.6450 \pm 0.0102$ | $0.5316 \pm 0.0190$ | $0.5587 \pm 0.0265$ | $0.8958 \pm 0.0108$ | $0.7856 \pm 0.0108$ | $0.6992 \pm 0.0090$ |
| **CondAdv** | $0.6437 \pm 0.0174$ | $0.5363 \pm 0.0227$ | $0.5571 \pm 0.0132$ | $0.8976 \pm 0.0030$ | $0.7898 \pm 0.0198$ | $0.6875 \pm 0.0130$ |
| **DANN** | $0.6480 \pm 0.0091$ | $0.5351 \pm 0.0162$ | $0.5679 \pm 0.0030$ | $0.8942 \pm 0.0120$ | $0.7852 \pm 0.0031$ | $0.6996 \pm 0.0071$ |
| **IRM** | $0.6479 \pm 0.0153$ | $0.5422 \pm 0.0101$ | $0.5750 \pm 0.0100$ | $0.8864 \pm 0.0084$ | $0.7821 \pm 0.0187$ | $0.7004 \pm 0.0192$ |
| **SagNet** | $0.6532 \pm 0.0083$ | $0.5508 \pm 0.0171$ | $0.5812 \pm 0.0251$ | $\underline{0.9024 \pm 0.0132}$ | $\underline{0.7990 \pm 0.0114}$ | $\mathbf{0.7053 \pm 0.0141}$ |
| **PCL** | $0.6588 \pm 0.0146$ | $\underline{0.5528 \pm 0.0156}$ | $\underline{0.5830 \pm 0.0131}$ | $0.8971 \pm 0.0165$ | $0.7909 \pm 0.0098$ | $0.7026 \pm 0.0253$ |
| **MLDG** | $0.6524 \pm 0.0178$ | $0.5476 \pm 0.0091$ | $0.5734 \pm 0.0148$ | $0.8890 \pm 0.0122$ | $0.7743 \pm 0.0102$ | $0.6904 \pm 0.0137$ |
| `ManyDG` | $\mathbf{0.6754 \pm 0.0079}$ | $\mathbf{0.5627 \pm 0.0066}$ | $\mathbf{0.6015 \pm 0.0120}$ | $\mathbf{0.9055 \pm 0.0054}$ | $0.7998 \pm 0.0124$ | $\underline{0.7015 \pm 0.0027}$ |

\* **Bold** for the best and underscore for the second best model. Our improvements are mostly significant tested in Appendix B.3.

### 4.3 RESULTS ON EHRs: DRUG RECOMMENDATION AND READMISSION PREDICTION

**EHR Databases** Then, we evaluate our model using electronic health records (EHRs) collected during patient hospital visits, which contain all types of clinical events, medical codes, and values. *MIMIC-III* (Johnson et al., 2016) is a benchmark EHR database, which contains more than 6,000 patients and 15,000 hospital visits, with 1,958 diagnosis types, 1,430 procedure types, 112 medication ATC-4 classes. We conduct the *drug recommendation* task on MIMIC-III. Following Shang et al. (2019); Yang et al. (2021a;b), the task is formulated as multi-label classification – predicting the prescription set for the current hospital visit based on longitudinal diagnosis and procedure medical codes. We adjust our decoder $\mathbf{p}_\psi(\cdot)$ to fit into the multi-label setting. Details can refer to Appendix A.4. The multi-center *eICU* dataset (Pollard et al., 2018) contains more than 140,000 patients' hospital visit records. We conduct a *readmission prediction* task on eICU, which is a binary classification task (Choi et al., 2020; Yang et al., 2022b), predicting the risk of being readmitted into ICU in the next 15 days based on the clinical activities during this ICU stay. We include five types of activities to form the heterogeneous sequence: diagnoses, lab tests, treatments, medications, and physical exams, extended from Choi et al. (2020), which only uses the first three types.

**Backbone and Metrics** For the multi-label drug recommendation task, we use a recent model GAMENet (Shang et al., 2019) as the backbone and employ the *Jaccard* coefficient, average area under the precision-recall curve (*Avg. AUPRC*) and *Avg. F1* over all drugs as metrics. Similarly, for the readmission prediction task, we consider the Transformer encoder (Vaswani et al., 2017) as backbone and employ the *AUPRC*, *F1* and Cohen's *Kappa* score as the metrics. Since both backbones do not have CNN structures, we drop SagNet in the following. Drug recommendation is not a multi-class classification, and we thus drop PCL as it does not fit into the setting.

**Results** The comparison results are shown in Table 2. Our `ManyDG` gives the best performance among all models, around 3.7% Jaccard and 1.2% AUPRC relative improvements against the best baseline on two tasks, respectively. We find that the meta-learning model MLDG gives poor performance. The reason might be that each patient has fewer samples in EHR databases compared to health monitoring datasets, and thus the MLDG episodic training strategy can be less effective. Other baseline models all have marginal improvements over the Base model.

Table 2: Result comparison on open EHR databases

| Models | Drug Recommendation | | | Readmission Prediction | | |
|---|---|---|---|---|---|---|
| | Jaccard | Avg. AUPRC | Avg. F1 | AUPRC | F1 | Kappa |
| **Base** | $0.4858 \pm 0.0139$ | $0.7440 \pm 0.0115$ | $0.6443 \pm 0.0076$ | $0.7040 \pm 0.0120$ | $0.6554 \pm 0.0089$ | $0.5210 \pm 0.0087$ |
| **CondAdv** | $\underline{0.4990 \pm 0.0144}$ | $\underline{0.7570 \pm 0.0052}$ | $0.6568 \pm 0.0131$ | $0.7047 \pm 0.0045$ | $\underline{0.6709 \pm 0.0035}$ | $0.5348 \pm 0.0135$ |
| **DANN** | $0.4886 \pm 0.0267$ | $0.7519 \pm 0.0137$ | $\underline{0.6595 \pm 0.0206}$ | $0.7152 \pm 0.0076$ | $0.6671 \pm 0.0051$ | $0.5309 \pm 0.0050$ |
| **IRM** | $0.4933 \pm 0.0083$ | $0.7537 \pm 0.0096$ | $0.6519 \pm 0.0056$ | $\underline{0.7174 \pm 0.0138}$ | $0.6614 \pm 0.0111$ | $\underline{0.5362 \pm 0.0114}$ |
| **PCL** | / | / | / | $0.7089 \pm 0.0143$ | $0.6575 \pm 0.0126$ | $0.5279 \pm 0.0274$ |
| **MLDG** | $0.4619 \pm 0.0100$ | $0.7051 \pm 0.0216$ | $0.6269 \pm 0.0239$ | $0.6701 \pm 0.0209$ | $0.6215 \pm 0.0256$ | $0.5178 \pm 0.0318$ |
| `ManyDG` | $\mathbf{0.5175 \pm 0.0130}$ | $\mathbf{0.7746 \pm 0.0035}$ | $\mathbf{0.6737 \pm 0.0141}$ | $\mathbf{0.7258 \pm 0.0132}$ | $\mathbf{0.6752 \pm 0.0025}$ | $\mathbf{0.5531 \pm 0.0144}$ |

### 4.4 VERIFYING THE ORTHOGONALITY OF EMBEDDING SUBSPACES

**Experimental Settings** This section provides quantitative evidence to show that the learned universal features $\mathbf{v}$ *stores domain and label information along approximately different dimensions*:

- First, we load the pre-trained embeddings from the $\mathbf{v}$, $\mathbf{v}_{\|\mathbf{z}}$ (equivalently $\mathbf{z}$), and $\mathbf{v}_{\perp\mathbf{z}}$ spaces;

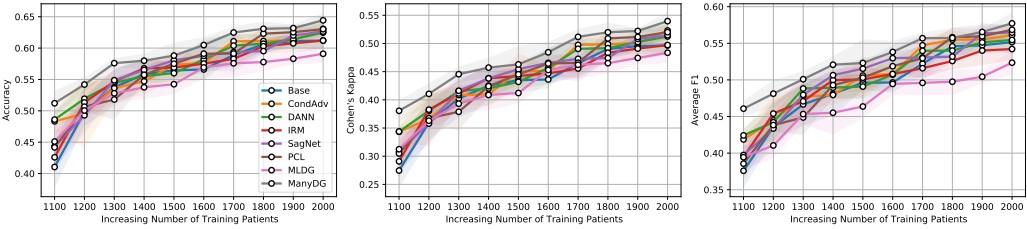

Figure 3: Continuous learning on new patients

- Then, following Shen et al. (2022), we learn a linear model (e.g., logistic regression) on the fixed embeddings to predict the domains and labels, formulated as multi-class classifications.
- The *cosine similarity of the learned weights* is reported for quantifying feature dimension overlaps.

To ensure that each dimension is comparable, we normalize $\mathbf{v}$, $\mathbf{z}$ and $\mathbf{v}_{\perp\mathbf{z}}$ dimension-wise by the standard deviation values before learning the linear weights. The results are shown in Table 3.

Recall that $\mathbf{z}$ is the learned domain representation, $\mathbf{v}$ is decomposed into $\mathbf{v}_{||\mathbf{z}}$ and $\mathbf{v}_{\perp\mathbf{z}}$ via orthogonal projection and they satisfy $\mathbf{v} = \mathbf{v}_{||\mathbf{z}} + \mathbf{v}_{\perp\mathbf{z}}$, where $\mathbf{v}_{||\mathbf{z}}$ ($\mathbf{v}_{\perp\mathbf{z}}$) is parallel (orthogonal) to $\mathbf{z}$. In Table 3, the first two rows give relatively higher similarity and imply the domain and label information are separated from $\mathbf{v}$ into $\mathbf{v}_{||\mathbf{z}}$ (equivalently $\mathbf{z}$) and $\mathbf{v}_{\perp\mathbf{z}}$, respectively. The third row has very low similarity (or even negative value), which indicates that the feature dimensions used for predicting (i) labels and (ii) domains are largely non-overlapped in $\mathbf{v}$. Note that the cosine similarity values from different tasks are not comparable. We provide more explanations in Appendix B.8.

Table 3: Cosine similarity of linear weights on $\mathbf{v}$, $\mathbf{v}_{||\mathbf{z}}$, $\mathbf{v}_{\perp\mathbf{z}}$

| Cosine Similarity | Seizure Detection | Sleep Staging | Drug Recom-mendation | Readmission Prediction | Average |
|---|---|---|---|---|---|
| `Weight`($\mathbf{v} \to$ labels) and `Weight`($\mathbf{v}_{\perp\mathbf{z}} \to$ labels) | 0.8789 | 0.8251 | 0.5714 | 0.4627 | $0.6845 \pm 0.1729$ |
| `Weight`($\mathbf{v} \to$ domains) and `Weight`($\mathbf{v}_{||\mathbf{z}} \to$ domains) | 0.2462 | 0.3273 | 0.1987 | 0.2111 | $0.2458 \pm 0.0510$ |
| `Weight`($\mathbf{v} \to$ labels) and `Weight`($\mathbf{v} \to$ domains) | 0.0182 | 0.0276 | 0.0445 | -0.0119 | $0.0196 \pm 0.0204$ |
| `Weight`($\mathbf{v} \to$ labels) and `Weight`($\mathbf{v}_{||\mathbf{z}} \to$ labels) | 0.1151 | 0.0773 | 0.0695 | 0.0524 | $0.0785 \pm 0.0229$ |

* `Weight`($\cdot$) denotes the learned linear weights. For example, `Weight`($\mathbf{v} \to$ labels) means that we train a linear model on $\mathbf{v}$ to predict the labels and then output the learned weights. Cosine similarity values are averaged over all classes in the training set.

## 4.5   CASE STUDIES: CONTINUOUS LEARNING ON NEW LABELED DATA

We simulate a realistic clinical setting in this section: continuous learning on newly available data. Another common setting is using insufficient labeled data, which is discussed in appendix B.1.

It is common in healthcare that new patient data is labeled gradually, and pre-trained predictive models will further optimize the accuracy based on these new annotations, instead of training on all existing data from scratch. We simulate this setting on the seizure detection task by sorting patients by their ID (smaller ID comes to the clinics first) in ascending order. Specifically, we pre-train each model on the first 1,000 patients. Then, we add 100 new patients at a step following the order. We fine-tune the models on these new patients and a small subset of previous data (we randomly select four samples from each previous patient and add them to the training set). Note that, without this small subset of existing data, all models will likely fail due to dramatic data shifts. The result of 10 consecutive steps is shown in Figure 3. Our model consistently performs the best during the continuous learning process. It is also interesting that MLDG improves slowly compared to other methods. The reason might be similar to what we have explained in Section 4.3 that the existing patients have only a few (e.g., 4) samples.

## 5   CONCLUSION

This paper proposes a novel many-domain generalization setting and develops a scalable model `ManyDG` for addressing the problem. We elaborate on our new designs in healthcare applications by assuming each patient is a separate domain with unique covariates. Our `ManyDG` method is built on a latent generative model and a factorized prediction model. It leverages mutual reconstruction and orthogonal projection to reduce the domain covariates effectively. Extensive evaluations on four healthcare tasks and two challenging case scenarios show our `ManyDG` has good generalizability.

## ACKNOWLEDGEMENTS

This work was supported by NSF award SCH-2205289, SCH-2014438, IIS-1838042, NIH award R01 1R01NS107291-01.

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

## A  DETAILS FOR NOTATIONS, IMPLEMENTATIONS AND DATASETS

### A.1  NOTATION TABLE

For clarity, we have attached a notation table here, summarizing all symbols used in the main paper. This paper uses the generic notation $\mathbf{x}$ to denote the input features. We use plain letters for scalars, such as $d$, $p$, $y$; boldface lowercase letters for vectors or vector-valued functions, e.g., $\mathbf{v}$, $\mathbf{z}$, $\mathbf{h}_\theta(\cdot)$, $\mathbf{q}_\phi(\cdot)$, $\mathbf{g}_\xi(\cdot)$, $\mathbf{p}_\psi(\cdot)$, and they have the same dimensionality; Euler script letters for sets, e.g., $\mathcal{D}$, $\mathcal{S}$.

Table 4: Notations used in `ManyDG`

| Symbols | Descriptions |
|---|---|
| $d \in \mathcal{D}$, $d' \in \mathcal{D}'$ | patient/domain set in training and test |
| $i \in \mathcal{S}^d$ | data sample set of patient/domain $d$ |
| $\mathbf{x}_i^d$, $y_i^d$ | the $i$-th (sample, class label) for patient/domain $d$ |
| $p(\cdot)$ | meta domain distribution |
| $\mathbf{z}$ | domain latent factor (representation) |
| $p(\cdot\|\mathbf{z}, y)$ | sample distribution conditioned on patient domain $\mathbf{z}$ and class label $y$ |
| $p(y\|\mathbf{x})$ | a generic notation for posterior of label probability given sample $\mathbf{x}$ |
| $p(\mathbf{z}\|\mathbf{x})$ | a generic notation for posterior of domain factor given sample $\mathbf{x}$ |
| $p(y\|\mathbf{x}, \mathbf{z})$ | a generic notation for posterior of label probability given sample $\mathbf{x}$ and domain $\mathbf{z}$ |
| $\mathbf{v}$ | the feature representation of $\mathbf{x}$ |
| $\hat{\mathbf{v}}$ | reconstructed feature representation of $\mathbf{x}$ |
| $\mathbf{v}_{\|\|\mathbf{z}}$, $\mathbf{v}_{\perp\mathbf{z}}$ | the component of $\mathbf{v}$ that is parallel to or orthogonal to $\mathbf{z}$ space |
| $\mathbf{h}_\theta(\cdot) : \mathbf{x} \mapsto \mathbf{v}$ | learnable feature extractor with parameter set $\theta$ |
| $\mathbf{q}_\phi(\cdot) : \mathbf{v} \mapsto \mathbf{z}$ | learnable encoder network for $p(\mathbf{z}\|\mathbf{x})$ (on top of $\mathbf{h}_\theta(\cdot)$) with parameter set $\phi$ |
| $\mathbf{g}_\xi(\cdot) : \mathbf{v}, \mathbf{z} \mapsto y$ | learnable predictor network for $p(y\|\mathbf{x}, \mathbf{z})$ (on top of $\mathbf{h}_\theta(\cdot)$) with parameter set $\xi$ |
| $\mathbf{p}_\psi(\cdot) : \mathbf{z}, y \mapsto \mathbf{v}$ | learnable decoder network for $p(\cdot\|\mathbf{z}, y)$ with parameter set $\psi$ |
| $\langle \cdot \rangle$ | vector inner product |
| $\|\cdot\|_{\mathcal{F}}$; $\|\cdot\|$ | vector norm induced by measure $\mathcal{F}$; $L_2$-norm |
| `Weight`$(\cdot)$ | the learned weight of a logistic regression task |

### A.2  IMPLEMENTING OUR DOUBLE DATA LOADER FOR TRAINING

Our method is trained end-to-end in a mini-batch fashion. We build a new data loader at each epoch. First, during each training epoch, we will randomly shuffle the data samples within each patient domain. Second, for building the data loader, we will fold each patient data list into two half-lists and append them to two global data lists respectively patient-by-patient. These two global lists have the same length and will be used to build the data loader. Upon using it, the data loader will output two data batches simultaneously (for the Siamese-type architecture), where the same indices correspond to the samples from the same patient. *At every epoch, only one pairing combination is used for the data from each patient, and the shuffle and re-build steps between epochs ensure that every data pair within the same patient can have the chance to be trained together with equal probability.* Our double data loader building procedure does not incur extra time consumption compared to the model training time. Our model's space and time complexity are asymptotically the same as training the Base model, shown in Appendix B.2.

### A.3  DETAILS IN IMPLEMENTING THE LABEL PREDICTOR AND DATA DECODER

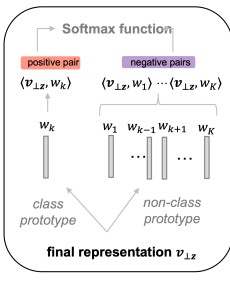

Figure 4: Predictor

**Prototype-base Predictor**  As mentioned in the main text, we use a fully connected (FC) layer without the bias term to be the predictor, which is essentially a parameter matrix $\mathbf{W}$. Each row in the parameter matrices $\{\mathbf{w}_k\}_{k=1}^K$ has the same dimensionality as $\mathbf{v}$ and can be viewed as the prototype for the class $k \in \{1, 2, \ldots, K\}$. We can construct one positive pair between the data representation and the corresponding class prototype and $(K-1)$ negative pairs with the prototypes from other classes (in Figure 4). The final label probability is given by the softmax activation form,

$$p(y = k|\mathbf{x}, \mathbf{z}) = \mathbf{g}_{\xi,k}(\mathbf{v}_{\perp\mathbf{z}}) = \frac{\exp(\langle \mathbf{w}_k, \mathbf{v}_{\perp\mathbf{z}}\rangle/\tau)}{\sum_i \exp(\langle \mathbf{w}_i, \mathbf{v}_{\perp\mathbf{z}}\rangle/\tau)}. \quad (16)$$

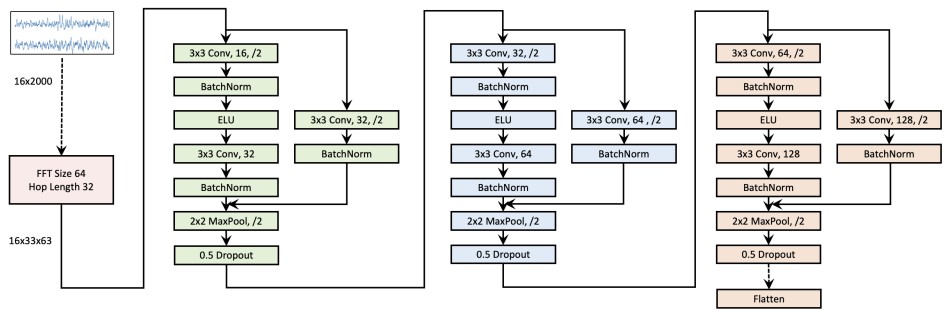

Figure 5: Shared backbone for seizure detection task

Here, $\langle \cdot, \cdot \rangle$ is the notation for vector inner product, and $\tau$ is the temperature (He et al., 2020). With the prototype-based design, our predictor is also amenable for use with recent supervised contrastive loss (Yao et al., 2022; Khosla et al., 2020); however, we leave this as future work.

**Prototype-base Data Decoder**  In the main text, we use a decoder network $\mathbf{p}_\psi(\cdot|\mathbf{z}, y)$ to reconstruct another sample from the same domain based on the domain factor $\mathbf{z}$ and the class label $y$. In the implementation, we do not directly input a combination of $\mathbf{z}$ and the categorical label $y$. For parameterizing the class label, we also use the prototype $\mathbf{w}_y$ as the class label information and concatenate it with the domain factor $\mathbf{z}$ for reconstruction.

### A.4 More information for datasets and implementation

**IIIC Seizure**  requested from (Jing et al., 2018; Ge et al., 2021; Jing & et al., 2023). We could not find the license, as stated in Jing et al. (2018) "the local IRB waived the requirement for informed consent for this retrospective analysis of EEG data".

We use 16-channel 10-second signals sampled at 200Hz. Thus, the raw data inputs are a matrix $16 \times 2000$, and the values are measurement amplitudes. We use *torch.stft* to implement the short-time Fourier transform (STFT) for obtaining the spectrogram. The FFT window size is 64, the hop length (overlapping window) is 32, and the imaginary and real parts are square-summed to be the energy density in the frequency domain. After STFT, the sample size is $16 \times 33 \times 63$. We further normalize the values and take the logarithm on one side of the spectrogram as the input to the backbone model. The data processing steps largely followed Jing et al. (2018). Seizure detection is a six-class classification problem, and we use some common combinations of hyperparameters: 128 as the batch size, 128 as the hidden representation size, 50 as the training epochs ($50 \times \dfrac{\text{size of an epoch}}{\text{size of an episode}}$ as the number of episodes for MLDG baseline, which follows MAML, the same setting for other datasets), $5 \times 10^{-4}$ as the learning rate with Adam optimizer and $1 \times 10^{-5}$ as the weight decay for all models. Our model uses $\tau = 0.5$ as the temperature (the same for other datasets). Other hyperparameters in baselines are mostly following their original default values. Some other variants of hyperparameter combinations have also been tested with the validation set (on the following three datasets, we also do the same procedures), which do not show significant differences. We report our backbone model in Figure 5.

The original dataset provides a vote count distribution over six labels as the targets in Seizure. We filter out the data that have fewer than five expert votes. We use the vote counts for calculating the following customized cross entropy loss (CEL) and use the majority label (that has the most significant vote) for calculating the evaluation metrics.

For calculating the supervised loss in Seizure, we use a customized version of cross entropy loss. Our customized CEL is a weighted average form of the standard CEL for the classification task. We employ the customized CEL form here, considering that for Seizure (and some other expert-annotated datasets), the labeling experts might have disagreements on the true label of ambiguous samples and thus usually resulting in a vote count distribution instead of a single label. The cus-

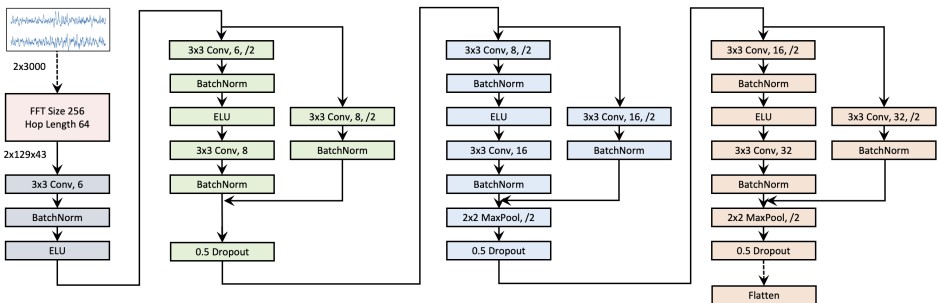

Figure 6: Shared backbone for sleep staging task

tomized CEL can capture more information (e.g., how hard the sample is) than the standard CEL in such cases. The customized CEL will reduce to the standard CEL if without labeling disagreements. We give the definition of our customized CEL below,

$$\mathcal{L}_4 = -\sum_{k \in \mathcal{S}} \frac{1}{|\mathcal{S}|} \log p(y = k|\mathbf{x}, \mathbf{z}). \tag{17}$$

where $\mathcal{S} \subset \{1, \cdots, K\}$ includes all class indices that have more votes than half of the maximum class votes. For example, if the votes for a sample are $[8, 0, 5, 3, 2, 1]$, then $\mathcal{S} = \{1, 3\}$ where $8$ and $5$ are viewed as valid votes, and other classes are considered minor classes, which will not be used for calculating the loss. For common classification task with single labels, $|\mathcal{S}| = 1$.

**Sleep-EDF** [2] (Kemp et al., 2000) We use the cassette portion. This dataset is under Open BSD 3.0 License[3]. It contains other Polysomnography (PSG) signals such as (horizontal) EOG, and submental chin EMG. We only use two EEG channels, Fpz-Cz and Pz-Oz, and a raw 30-second sample has size $2 \times 3000$. We also do STFT with an FFT window size of 256, hop length of 64 and normalize and logarithm operations on one side of the spectrogram. The data processing step[4] largely follows Yang et al. (2021c; 2022a). The final size of a sample is $2 \times 129 \times 43$. We integrated the data cleaning and processing pipeline into PyHealth Yang et al. (2022b). We select combinations of hyperparameters: 256 as the batch size, 128 as the hidden representation size, 50 as the training epochs, $5 \times 10^{-4}$ as the larning rate with Adam optimizer, and $1 \times 10^{-5}$ as the weight decay for all models. The backbone model is reported in Figure 6.

**MIMIC-III** [5] (Johnson et al., 2016) This dataset is under PhysioNet Credentialed Health Data License 1.5.0[6]. The dataset is publicly available with patients who stayed in intensive care unit (ICU) in the Beth Israel Deaconess Medical Center for over 11 years. It consists of 50,206 medical encounter records. Following the preprocessing step from Yang et al. (2021a;b); Shang et al. (2019). We filter out the patients with only one visit. Diagnosis, procedure and medication information is extracted in "DIAGNOSES_ICD.csv", "PROCEDURES_ICD.csv" and "PRESCRIPTIONS.csv" from original MIMIC database. We merge these three sources by patient id and visit id. After the merging, diagnosis and procedure are ICD-9 coded, and they will be transformed into multi-hot vectors before training. There are in total 6,350 patients, 15,032 visits, 1,958 types of diagnoses, 1,430 types of procedures, and 112 ATC-4 coded medications. For each visit, we use the diagnosis and procedure information (two ICD code sets) and previous visit information (two ICD code sets and one ATC code set per visit) to predict the current medication set, which is a sequential multi-label prediction task. Later on, we integrate the MIMIC-III processing pipeline into PyHealth[7] Yang et al. (2022b). In this task, we use binary cross entropy loss for each medication, following Yang et al. (2021a;b); Shang et al. (2019) and then aggregate the loss. The backbone model is

---

[2]https://www.physionet.org/content/sleep-edfx/1.0.0/

[3]https://www.physionet.org/content/sleep-edfx/view-license/1.0.0/

[4]https://github.com/ycq091044/ContraWR

[5]https://physionet.org/content/mimiciii/1.4/

[6]https://physionet.org/content/mimiciii/view-license/1.4/

[7]https://pyhealth.readthedocs.io

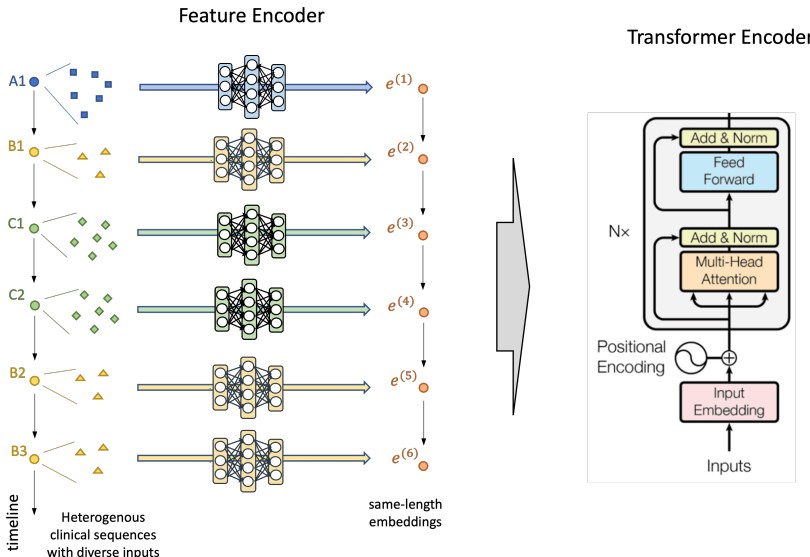

Figure 7: Shared backbone for the readmission prediction task. Different colored circles mean different event types and different feature inputs.

borrowed from a recent paper Shang et al. (2019) with the same model hyperparameter set. For other hyperparameters, we use 64 as the batch size, 64 as hidden representation, 100 as the training epochs, $1 \times 10^{-3}$ as the learning rate with Adam optimizer and $1 \times 10^{-5}$ as the weight decay for all models. We adjust the conditional mutual reconstruction part of our model by using the average of class prototypes as the label information for this setting.

**eICU** [8] (Pollard et al., 2018) This dataset is under PhysioNet Credentialed Health Data License 1.5.0[9]. This data covers patients admitted to critical care units from 2014 to 2015. We define each encounter as identified by *patientunitstayid* in the data tables. Data processing step largely follows Choi et al. (2020). We consider the following five event categories and the features: (i) diagnosis: diagnosis string features (in total 3,845 types) and the ICD-9 codes (in total 1,195 types); (ii) lab test: lab test types (in total 158 types) and the corresponding numeric results; (iii) medication: medication names (in total 291 types), prescription and the pick list frequency with which the drug is taken; (iv) physical exam: the system root path of the physical exam (in total 462 types), which indicates the types; (v) treatment: the path of the treatment (in total 2,695 types). The "offset" time in raw data tables is used as the timestamp for each event since they are offset with respect to the admission time. For each encounter, we can collect a heterogeneous event sequence following time order for predicting the risk of re-admitted into ICU (re-hospitalized) within next 15 days (Choi et al., 2020). We integrated the data cleaning and processing pipeline into PyHealth Yang et al. (2022b). In this task, we learn two class prototypes and finally feed the first column of the output softmax logits for the binary cross entropy loss (which only takes *batch_size* × *1* format as input in *torch.nn.functional.binary_cross_entropy_loss*). For hyperparameters, we use 256 as the batch size, 128 as hidden representation, 50 as the training epochs, $5 \times 10^{-4}$ as the learning rate with Adam optimizer, and $1 \times 10^{-5}$ as the weight decay for all models. The backbone model is based on 3-layer Transformer encoder blocks, before which we first use separate encoders for encoding different types of events into the same-length embeddings. We show an illustration in Figure 7.

We report the label distribution information in Table 5.

---

[8]https://physionet.org/content/eicu-crd/2.0/
[9]https://physionet.org/content/eicu-crd/view-license/2.0/

Table 5: Label information of four datasets

| Dataset | Task | Label Distribution |
|---------|------|--------------------|
| Seizure | multi-class | OTHER: 26.4%, ESZ: 3.7%, LPD: 19.7%, GPD: 24.2%, LRDA: 14.4%, GRDA: 11.7% |
| Sleep-EDF | multi-class | Wake: 68.8%, N1: 5.2%, N2: 16.6%, N3: 3.2%, REM: 6.2% |
| MIMIC-III | multi-label | max # of med. is 64, avg # of med. is 11.65 |
| eICU | binary | hospitalized: 83.78%, non-hospitalized: 16.22% |

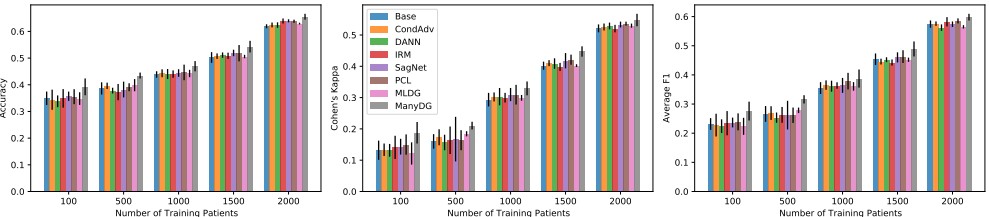

Figure 8: Varying number of training patients

# B  ADDITIONAL EXPERIMENTAL RESULTS

We present additional experiments in this appendix section due to space limitations.

## B.1  BETTER GENERALIZATION ABILITY ON SMALL TRAINING DATA

**Train on Small Data.**  Here, we mimic the critical situation where the labeling cost is very high (such as seizure detection (Jing et al., 2018)) and only limited training data is available. We rely on EEG seizure detection tasks and evaluate all models on insufficient labeled data. We limit the training number of patients to 2,000, 1,500, 1,000, 500, 100 and use the same test group for evaluation. The results are reported in Figure 8, which suggests that: (i) the performance of all models deteriorates with less labeled data; (ii) our ManyDG consistently outperforms the baselines.

## B.2  RUNNING TIME COMPARISON

This section compares the running time of all models in four healthcare tasks. As mentioned before, all models use the same backbone and batch size. When recording the running time, we duplicated the environment mentioned in Section 4.1, stopped other programs, and ran all the models one by one on one GPU. We record the first 23 epochs of all models and drop the first three epochs (since they might be unstable). We report the mean time cost per epoch in Table. MLDG uses an episodic training strategy, different from epoch-based training. Thus, we calculate the *equivalent running time* for MLDG, which is that we first average the episode time by the sample size and then multiply the per-sample time by the overall training data size.

We report the results in Table 6, which shows our method is as efficient as the Base model in each task. CondAdv and DANN are two adversarial training models which rely on an additional domain discriminator. In our cases with patient-induced domains, we have more domains to deal with (than previous domain generalization settings). Thus, their discriminator can be less effective and will cost more time.

Table 6: Running time comparison (seconds per epoch)

| Model | Seizure Detection | Sleep Staging | Drug Recommendation | Readmission Prediction |
|-------|-------------------|---------------|---------------------|------------------------|
| **Base** | $7.128 \pm 0.4000$ | $17.91 \pm 0.1885$ | $3.306 \pm 0.0219$ | $541.4 \pm 12.36$ |
| **CondAdv** | $19.95 \pm 0.1673$ | $20.60 \pm 0.1498$ | $3.472 \pm 0.0248$ | $580.1 \pm 16.87$ |
| **DANN** | $11.71 \pm 0.3037$ | $18.29 \pm 1.0210$ | $3.518 \pm 0.0256$ | $585.3 \pm 8.282$ |
| **IRM** | $7.598 \pm 0.3639$ | $19.23 \pm 0.5725$ | $3.416 \pm 0.0022$ | $558.2 \pm 4.022$ |
| **SagNet** | $12.75 \pm 0.4110$ | $35.57 \pm 0.0186$ | / | / |
| **PCL** | $8.743 \pm 0.4869$ | $21.70 \pm 0.3514$ | / | $577.9 \pm 10.24$ |
| **MLDG (equivalent)** | $13.87 \pm 0.4965$ | $31.89 \pm 0.2578$ | $4.210 \pm 0.0448$ | $663.6 \pm 14.05$ |
| **ManyDG** | $7.996 \pm 0.2862$ | $19.23 \pm 0.2744$ | $3.462 \pm 0.0648$ | $558.1 \pm 14.27$ |

## B.3 STATISTICAL TESTING WITH P-VALUES

We also conduct T-tests on the results in Section 4.2 and Section 4.3 and calculated the p-values in the following Table 7 and Table 8. Commonly, a p-value smaller than 0.05 would be considered significant. We use **bold** font for p-values that is larger than 0.05. We can conclude that our performance gain is significant over the baselines in most of the cases.

Table 7: p-values under T-test on biosignal classification tasks

| Comparison | Seizure Detection | | | Sleep Staging | | |
| --- | --- | --- | --- | --- | --- | --- |
| | Accuracy | Kappa | Avg. F1 | Accuracy | Kappa | Avg. F1 |
| **Base vs ManyDG** | 1.8277e-04 | 2.3857e-03 | 3.1177e-03 | 3.9742e-02 | 3.1450e-02 | **2.7878e-01** |
| **CondAdv vs ManyDG** | 1.6107e-03 | 1.1745e-02 | 1.2661e-04 | 6.3333e-03 | **1.5790e-01** | 1.4948e-02 |
| **DANN vs ManyDG** | 2.3134e-04 | 2.1345e-03 | 6.9548e-05 | 3.2046e-02 | 1.0646e-02 | **2.7459e-01** |
| **IRM vs ManyDG** | 1.9953e-03 | 1.4039e-03 | 1.4164e-03 | 6.9382e-04 | 4.2033e-02 | **4.4536e-01** |
| **SagNet vs ManyDG** | 6.4119e-04 | 7.1616e-02 | **5.2789e-02** | 3.0083e-01 | **4.5421e-01** | **7.3663e-01** |
| **PCL vs ManyDG** | 1.8472e-02 | 9.1054e-02 | 1.5726e-02 | **1.3048e-01** | 9.8423e-02 | **5.4170e-01** |
| **MLDG vs ManyDG** | 9.1763e-03 | 4.9798e-03 | 3.0785e-03 | 7.4243e-03 | 2.0582e-03 | 4.1056e-02 |

Table 8: p-values under T-test on EHR classification tasks

| Comparison | Drug Recommendation | | | Readmission Prediction | | |
| --- | --- | --- | --- | --- | --- | --- |
| | Jaccard | Avg. AUPRC | Avg. F1 | AUPRC | F1 | Kappa |
| **Base vs ManyDG** | 1.5738e-03 | 1.0867e-04 | 8.9086e-04 | 7.8500e-03 | 3.4105e-04 | 7.0435e-04 |
| **CondAdv vs ManyDG** | 2.2134e-02 | 5.5230e-05 | 2.9717e-02 | 2.6847e-03 | 1.8490e-02 | 2.4544e-02 |
| **DANN vs ManyDG** | 2.0507e-02 | 1.9382e-03 | **9.6398e-02** | **6.0040e-02** | 3.6720e-03 | 3.2901e-03 |
| **IRM vs ManyDG** | 2.2013e-03 | 4.5726e-04 | 3.5304e-03 | **1.5173e-01** | 8.1280e-03 | 2.5219e-02 |
| **PCL vs ManyDG** | / | / | / | 4.3115e-03 | 4.3811e-03 | 3.8113e-02 |
| **MLDG vs ManyDG** | 1.4382e-05 | 2.3040e-05 | 1.4652e-03 | 2.4545e-04 | 4.0163e-04 | 1.7682e-02 |

## B.4 COSINE SIMILARITY ON DOMAIN REPRESENTATIONS

In this section, we load the pre-trained embedding $\mathbf{z}$ from four healthcare tasks. To give more insights into the learned domain factor $\mathbf{z}$, we compute the cosine similarity of the estimated $\mathbf{z}$ within the same domains and the cosine similarity of $\mathbf{z}$ cross domains. We average the computed cosine similarity values over each embedding pair and show the results in Table 9.

Table 9: Cosine similarity of within-domain and cross-domain factors $\mathbf{z}$

| Cosine Similarity | Seizure Detection | Sleep Staging | Drug Recom- mendation | Readmission Prediction | Average |
| --- | --- | --- | --- | --- | --- |
| Avg. of within-domain $\mathbf{z}$ | 0.9789 | 0.9672 | 0.9993 | 0.9619 | $0.9768 \pm 0.0144$ |
| Avg. of cross-domain $\mathbf{z}$ | 0.9685 | 0.9584 | 0.9965 | 0.9523 | $0.9689 \pm 0.0169$ |

It is interesting that both the within-domain and cross-domain similarity are pretty high, e.g., larger than 0.95. The second finding is that within-domain similarity is larger than cross-domain similarity (though it seems insignificant, we will explain why this phenomenon is normal and our design is however useful). The reason is that we only enforce the similarity of $\mathbf{z}$ from the same domain as one objective $\mathcal{L}_{\text{sim}}$ but never enforce the dissimilarity of $\mathbf{z}$ from different domains (if we did, then the values in second row of the table are expected to decrease).

A very similar phenomenon has been observed in Grill et al. (2020) from the self-supervised contrastive learning area. Before the proposal of Grill et al. (2020), researchers will simultaneously enforce the similarity of positive samples and the dissimilarity of negative samples He et al. (2020); Chen et al. (2020) for learning unsupervised features invariant to data augmentations. However, Grill et al. (2020) claims that enforcing the similarity of positive pairs along is empirically good enough for learning unsupervised features. In the implementation, the cosine similarity of positive pairs is learned to approach 1.0, and the negative pairs will chase after to achieve a slightly lower score (often times higher than 0.95). They show better results on ImageNet against previous approaches He et al. (2020); Chen et al. (2020) (that enforces both similarity and dissimilarity).

Our method empirically also works well on four datasets. Further discussion on this topic is beyond the main scope of the paper. We will explore this direction for future work.

### B.5 ABLATION STUDY ON OBJECTIVES AND THEIR COMBINATIONS

**Ablation Study on Combinations of Objectives**    First, we design ablation studies to test the effectiveness of each objective. Recall that, we have used the following final loss in the main text:

$$\mathcal{L}_{\text{final}} = \mathcal{L}_{\text{sup}} + \mathcal{L}_{\text{MMD}} + \mathcal{L}_{\text{rec}} + \mathcal{L}_{\text{sim}}. \tag{18}$$

Here, $\mathcal{L}_{\text{sup}}$ is the supervised cross entropy loss, $\mathcal{L}_{\text{MMD}}$ is the loss to enforce $\mathbf{v}$ and $\mathbf{z}$ to be in the same space, $\mathcal{L}_{\text{rec}}$ is the mutual reconstruction loss between two data samples, and $\mathcal{L}_{\text{sim}}$ enforces the similarity of two latent factors from the same patient.

In this section, we test the following loss combinations on the seizure detection task: **(i) Case 1**: Only $\mathcal{L}_{\text{sup}}$; **(ii) Case 2**: $\mathcal{L}_{\text{sup}}$ and $\mathcal{L}_{\text{MMD}}$; **(iii) Case 3**: $\mathcal{L}_{\text{sup}}$ and $\mathcal{L}_{\text{rec}}$; **(iv) Case 4**: $\mathcal{L}_{\text{sup}}$, $\mathcal{L}_{\text{MMD}}$ and $\mathcal{L}_{\text{rec}}$. We show the final performance results and the learning curve of each objective in Figure 9. The results prove the necessity of all our objectives. We provide discussions and insights on the effectiveness of individual objectives:

- With different objectives, the overall losses all decrease, which means the model trains successfully in each case.

- **In case 1 (only $\mathcal{L}_{sup}$, no $\mathcal{L}_{MMD}$, $\mathcal{L}_{rec}$, $\mathcal{L}_{sim}$)**, there is no interaction between two sides of the Siamese architecture. **Our method performs similarly to the Base model.** Reflected on the loss curves: $\mathcal{L}_{sim} = -0.17$ (domain factors are not similar, ideal value is -1.0); $\mathcal{L}_{rec} = 0$ (no reconstruction effect, ideal value is -2.0); $\mathcal{L}_{MMD} = 2.0$ (embedding spaces are not aligned, ideal value is 0).

- **In case 2 (only $\mathcal{L}_{sup}$, $\mathcal{L}_{MMD}$, no $\mathcal{L}_{rec}$, $\mathcal{L}_{sim}$):** There is no interaction between two sides of the Siamese architecture. **Our method performs similarly to the Base model.** However, the MMD objective improves the orthogonal projection, which explains the slight improvements over **case 1**. Reflected on the loss curves: $\mathcal{L}_{sim} = -0.45$ (domain factors are somewhat similar, ideal value is -1.0); $\mathcal{L}_{rec} = 0$ (no reconstruction effect, ideal value is -2.0); $\mathcal{L}_{MMD} = 0$ (embedding spaces are aligned, ideal value is 0).

- **In case 3 (only $\mathcal{L}_{sup}$, $\mathcal{L}_{rec}$, no $\mathcal{L}_{MMD}$, $\mathcal{L}_{sim}$): The reconstruction objective ensures the interactions between two-sided pipeline, which is essential for learning the domain and domain-invariance representations.** We find that without $\mathcal{L}_{MMD}$, our model can learn to align the $\mathbf{v}$ and $\mathbf{z}$ embedding spaces and use orthogonal projection for prediction. According to the bar chart in Figure 9, \*\*the $\mathcal{L}_{rec}$ brings the most performance gains\*\* compared to other objectives (exclude $\mathcal{L}_{sup}$). Reflected on the loss curves: $\mathcal{L}_{sim} = -0.45$ (domain factors are somewhat similar, ideal value is -1.0); $\mathcal{L}_{rec} = -1.4$ (reconstruction effect, ideal value is -2.0); $\mathcal{L}_{MMD} = 0.15$ (embedding spaces are nearly aligned, ideal value is 0).

- **In case 4 ($\mathcal{L}_{sup}$, $\mathcal{L}_{rec}$, $\mathcal{L}_{MMD}$, no $\mathcal{L}_{sim}$):** Adding $\mathcal{L}_{MMD}$ can ensure the perfect alignment of the $\mathbf{v}$ and $\mathbf{z}$ embedding spaces and brings further improvements over **case 3**. Reflected on the loss curves: $\mathcal{L}_{sim} = -0.45$ (domain factors are somewhat similar, ideal value is -1.0); $\mathcal{L}_{rec} = -1.4$ (reconstruction effect, ideal value is -2.0); $\mathcal{L}_{MMD} = 0$ (embedding spaces are aligned, ideal value is 0).

- Adding $\mathcal{L}_{sim}$ for **case 4** leads to the final objective in our method. $\mathcal{L}_{sim}$ maximizes the similarity of the domain factors and brings further accuracy improvements.

In summary, $\mathcal{L}_{sup}$ brings the label supervision (the most important one), $\mathcal{L}_{rec}$ brings additional information (tells the model that two samples are from the same domain), $\mathcal{L}_{MMD}$ improves the orthogonal projection (for better disentanglement), and $\mathcal{L}_{sim}$ improves the learned domain factor (for better orthogonal projection).

**Ablation Study on Combinations of Weights**    Next, on the seizure detection task, we add weights as hyperparameters $\lambda_1, \lambda_2, \lambda_3 > 0$ to combine our proposed objectives,

$$\mathcal{L}_{\text{final}} = \mathcal{L}_{\text{sup}} + \lambda_1 \cdot \mathcal{L}_{\text{MMD}} + \lambda_2 \cdot \mathcal{L}_{\text{rec}} + \lambda_3 \cdot \mathcal{L}_{\text{sim}}. \tag{19}$$

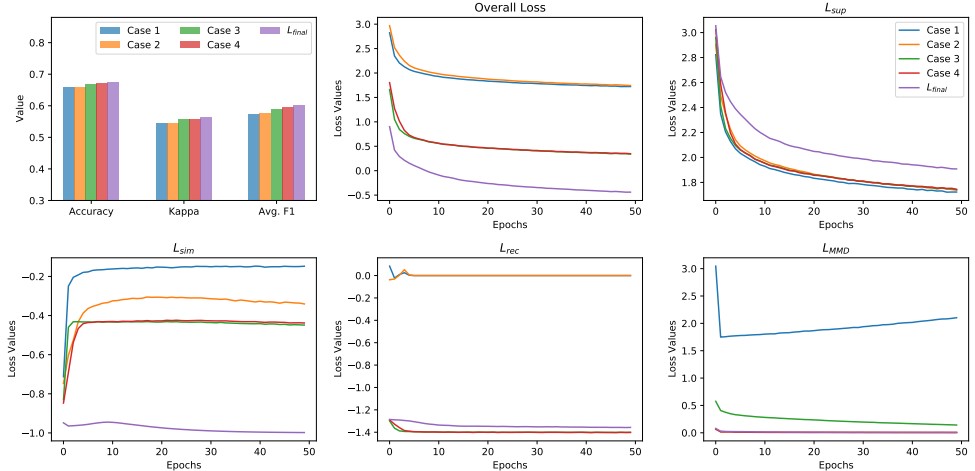

Figure 9: Results of ablation studies on different objective combinations

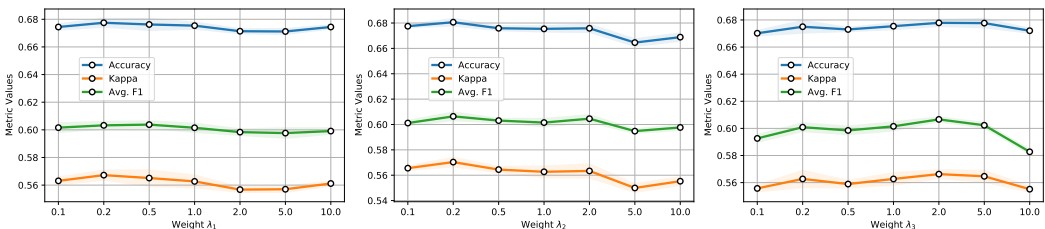

Figure 10: Results on individual weight variation

By default, we use $\lambda_1 = \lambda_2 = \lambda_3 = 1.0$ in the experiments of main text. This section first tests different variations of $\lambda_1, \lambda_2, \lambda_3$ individually. The results are shown in Figure 10. When increasing one of the weights from 0.1 to 10.0 and fix the others, we find that all three metrics will follow a similar pattern: first become better and then slightly decreases. From the figure, we can conclude that using $\lambda_1 = \lambda_2 = \lambda_3 = 1.0$ is indeed a decent choice without using exhaustive search. However, there are minor variations in performance, and apparently we can find a (slightly) better combination than default.

Based on the individual performance, we guess that $\lambda_1 = 0.2, \lambda_2 = 0.2, \lambda_3 = 2.0$ might be a better combination from a greedy perspective. Thus, we train the model again on this combination and find that it gives marginal improvements over our default choice, shown in Table 10.

Table 10: Comparison with greedy weight combination

| Weights | Accuracy | Kappa | Avg. F1 |
|---|---|---|---|
| $\lambda_1 = \lambda_2 = \lambda_3 = 1.0$ | $0.6754 \pm 0.0079$ | $0.5627 \pm 0.0066$ | $0.6015 \pm 0.0120$ |
| $\lambda_1 = 0.2, \lambda_2 = 0.2, \lambda_3 = 2.0$ | $0.6785 \pm 0.0045$ | $0.5654 \pm 0.0062$ | $0.6031 \pm 0.0089$ |

### B.6 SCATTER PLOT OF $\mathbf{v}_{\perp\mathbf{z}}, \mathbf{v}_{||\mathbf{z}}$

To provide more insights on the learned embeddings, we plot the L2-norm statistics of $\mathbf{v}_{\perp\mathbf{z}}, \mathbf{v}_{||\mathbf{z}}$ on four tasks for both training and test. Specifically, we draw a scatter plot using L2-norm of $\mathbf{v}_{\perp\mathbf{z}}$ as x-axis and L2-norm of $\mathbf{v}_{||\mathbf{z}}$ as y-axis.

For IIIC seizure dataset, we randomly select 25 batches (3,200 samples) from the training set and select 25 batches (3,200 samples) from the test set. For Sleep-EDF sleep staging dataset, we randomly select 25 batches (3,200 samples) from the training set and 25 batches (3,200 samples) from

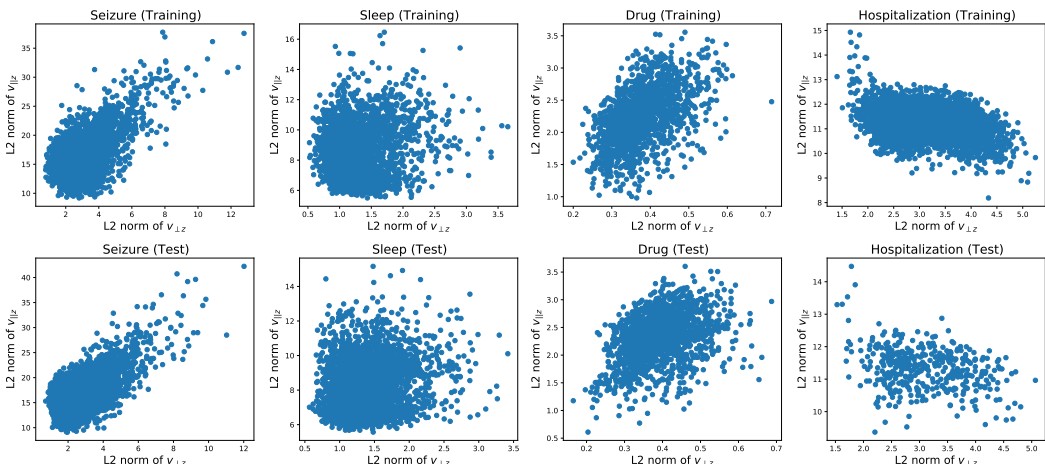

Figure 11: Scatter plot of $\mathbf{v}_{\perp\mathbf{z}}$ and $\mathbf{v}_{||\mathbf{z}}$

the test set. For MIMIC-III drug recommendation dataset, we randomly select 25 batches (1,600 samples) from the training set and 25 batches (1,600 samples) from the test set. For eICU hospital readmission prediction dataset, we randomly select 25 batches (1,600 samples) from the training set and 10 batches (640 samples) from the test set. The results are shown in Figure 11.

**Result Analysis** From Figure 11, we can roughly come up with three findings: (i) The scatter plot distributions for training and test look similar on all datasets and tasks, which means our approach is well-trained and generaliable (at least on the norm space); (ii) The magnitude (norm) of $\mathbf{v}_{||\mathbf{z}}$ is generally larger than $\mathbf{v}_{\perp\mathbf{z}}$ on all datasets, which means the domain information $\mathbf{v}_{||\mathbf{z}}$ is indeed a large component on the extracted features $\mathbf{v}$. This finding again verifies the necessity of removing the domain information during the prediction; (iii) The correlations between norms of $\mathbf{v}_{||\mathbf{z}}$ and $\mathbf{v}_{\perp\mathbf{z}}$ are slightly different across different tasks. In the IIIC seizure prediction task and drug MIMIC recommendation task, their norms seem a bit positively correlated. In Sleep-EDF sleep staging task, their norms seem uncorrelated. In eICU hospitalization prediction task, their norms seem a bit negative correlated. We conclude that the norm correlations are highly dependent on the dataset characteristics.

### B.7 PERFORMANCE COMPARISON WITH SIMPLE DECOMPOSITION MODEL

In this section, we compare our model performance with two simple decomposition models. They have the same architectural designs with different objective functions. We describe the architecture:

- We assume the model has a **feature encoder** from $\mathbf{x} \rightarrow \mathbf{v}$, and $\mathbf{v}$ contains both **domain information** and **domain-invariant label information**.

- We assume the model has a **domain information encoder** that maps $\mathbf{v}$ into a domain representation $\mathbf{z}$ and has a **label information encoder** that maps $\mathbf{v}$ into a label representation $\mathbf{u}$.

- There are four loss objectives, which (i) **(supervised loss)** minimizes the supervised prediction loss with a final prediction layer on $\mathbf{u}$, similarity on $\mathbf{u}$ as well; (ii) **(reconstruction loss)** minimizes the discrepancy between $\mathbf{v}$ and the reconstructed $\hat{\mathbf{v}}$ from $\mathbf{z}, \mathbf{u}$, similarity for $\mathbf{v}'$ and $\hat{\mathbf{v}}'$ from $\mathbf{z}', \mathbf{u}'$; (iii) **(domain similarity loss)** maximizes the similarity of $\mathbf{z}$ and $\mathbf{z}'$ between two samples of the same patient domain; (iv) **(orthogonality loss)** enforces the orthogonality of $\mathbf{z}, \mathbf{u}$ and the orthogonality of $\mathbf{z}', \mathbf{u}'$.

For the **first simple decomposition (SimD1) model**, we follow this domain adaptation work (Bousmalis et al., 2016), which also learns the domain-specific information and domain-shared information separately by two neural networks. It uses (i) cross-entropy loss as the **supervised loss**; (ii) mean square error (MSE) as the **reconstruction loss**; (iii) scale-invariant MSE (Eigen et al., 2014) as the **domain similarity loss**; (iv) Frobenius norm as the **orthogonality loss**.

For the **second simple decomposition (SimD2) model**, we use the loss design from our approach and use (i) cross-entropy loss as the **supervised loss**; (ii) normalized cosine similarity as the **reconstruction loss**; (iii) normalized cosine similarity as the **domain similarity loss**; (iv) Frobenius norm as the **orthogonality loss**.

**Result Analysis**  We show the performance compared of these two simple decomposition models with our proposed `ManyDG` in Table 11 and Table 12. In summary, our `ManyDG` outperforms two simple decomposition models significantly, which shows the effectiveness of our mutual reconstruction and orthogonal projection steps. The SimD2 model also performs better than SimD1 model consistently. The only differences between them are the metrics used for each objective design. We find that SimD1 needs to carefully find the hyperparameter weights for balancing different objectives, while the objectives in SimD2 are all well-scaled and can be used together without weights. Though we have selected decent weight combinations for SimD1, it is still inferior to SimD2.

Table 11: Result comparison on health monitoring datasets

| Models | Seizure Detection | | | Sleep Staging | | |
|---|---|---|---|---|---|---|
| | Accuracy | Kappa | Avg. F1 | Accuracy | Kappa | Avg. F1 |
| **SimD1** | $0.5954 \pm 0.0127$ | $0.4792 \pm 0.0197$ | $0.4998 \pm 0.0242$ | $0.8862 \pm 0.0015$ | $0.7855 \pm 0.0149$ | $0.6962 \pm 0.0025$ |
| **SimD2** | $0.6499 \pm 0.0153$ | $0.5485 \pm 0.0065$ | $0.5686 \pm 0.0092$ | $0.9017 \pm 0.0073$ | $0.7978 \pm 0.0205$ | $0.6989 \pm 0.0141$ |
| **ManyDG** | $\mathbf{0.6754 \pm 0.0079}$ | $\mathbf{0.5627 \pm 0.0066}$ | $\mathbf{0.6015 \pm 0.0120}$ | $\mathbf{0.9055 \pm 0.0054}$ | $\mathbf{0.7998 \pm 0.0124}$ | $\mathbf{0.7015 \pm 0.0027}$ |

Table 12: Result comparison on open EHR databases

| Models | Drug Recommendation | | | Readmission Prediction | | |
|---|---|---|---|---|---|---|
| | Jaccard | Avg. AUPRC | Avg. F1 | AUPRC | F1 | Kappa |
| **SimD1** | $0.4894 \pm 0.0087$ | $0.7472 \pm 0.0065$ | $0.6482 \pm 0.0192$ | $0.6985 \pm 0.0153$ | $0.6368 \pm 0.0026$ | $0.5079 \pm 0.0118$ |
| **SimD2** | $0.4901 \pm 0.0091$ | $0.7536 \pm 0.0115$ | $0.6534 \pm 0.0127$ | $0.7047 \pm 0.0105$ | $0.6584 \pm 0.0048$ | $0.5260 \pm 0.0103$ |
| **ManyDG** | $\mathbf{0.5175 \pm 0.0130}$ | $\mathbf{0.7746 \pm 0.0035}$ | $\mathbf{0.6737 \pm 0.0141}$ | $\mathbf{0.7258 \pm 0.0132}$ | $\mathbf{0.6752 \pm 0.0025}$ | $\mathbf{0.5531 \pm 0.0144}$ |

## B.8    More explanations on the learned linear weights

In this section, we discuss the relatively low cosine similarity in Table 3. Compared to the first row (predicting the labels, e.g., 6-class classification in seizure detection), the second row corresponds to predicting one of the many domains (e.g., 2,702-class classification in seizure detection). Using the linear prediction model, the performances of the second row are naturally not as good as the first row. As a result, the learned coefficients in the second row are less similar than those learned in the first row. This may also explain why sleep staging has the highest similarity in the second row (since this task only has 78 domains).

