# OpenReview forum: "ManyDG: Many-domain Generalization for Healthcare Applications"
_ICLR.cc/2023/Conference — ICLR 2023 poster_

### Official Review · Reviewer_TUM8 · 2022-10-25

**Confidence:** 3
**Correctness:** 3
**Technical Novelty And Significance:** 3
**Empirical Novelty And Significance:** 3
**Recommendation:** 8

**Clarity, Quality, Novelty And Reproducibility:**

Paper is written in an understandable way, the approach seems novel and fairly well evaluated. Datasets are publicly available and details seems sufficient for reproducibility.

**Strength And Weaknesses:**

Strengths
As mentioned, compelling empirical evaluation, with favorable results against relevant contemporary approaches.
The architecture appears fairly novel, and scaling up the number of domains seems like a very relevant direction. To add just one comment, “many-domain” prefix does not do justice to the real scale of the tackled problem, as previous approaches also tackled many domains (eg six in Li et al. (2020)). To borrow a term from “multi-label classification” area, thousands of labels there would make the problem “extreme multi-label classification”, so maybe “extreme multi-domain” would fit here too.

Weakness
Ablation studied impact on loss function pattern only, but have not provided insight on how that impacts supervised task performance downstream.
In the “Remark for mutual reconstruction”, stating that z would not be incentivised to preserve label information requires a bit more support.


**Summary Of The Paper:**

Paper introduces a many-domain generalization approach, a siamese-type architecture that takes in paired samples from the same domain (e.g. patient) which learns sample as well as domain embedding and in the end predicts the label. The architecture is trained through optimization of a 4-part objective which aims to 1) mutually reconstruct the samples, 2) make domain representation similar, 3) try to align the sample and domain embeddings and 4) get accurate predictions. Extensive evaluation was performed on 4 datasets, plus two studies of effects of small datasets and continuous learning. Framework shows favorable results compared against several relevant competitors.

**Summary Of The Review:**

Based on the perceived novelty of the framework and thorough empirical evaluation with compelling results, I am on a positive side wrt this paper.

---

> ### Author Response · Authors · 2022-11-17
> **Response to Reviewer TUM8 (Part I)**
>
> We thank the reviewer for your appreciation and constructive comments.
>
> We have uploaded a revision and used blue to mark the new changes. Yes, the analogy of “multi-label classification” vs. “extreme multi-label classification” makes sense. We will consider using the term “extreme multi-domain generalization” in the final version.
>
> ---
>
> **Q1: Ablation studied the impact on loss function pattern only but has not provided insight on how that impacts supervised task performance downstream.**
>
> Thanks. We have added the following discussion after the ablation study (updated the revision):
>
> - With different objectives, the overall losses all decrease.
> - **In case 1 (only $\mathcal{L}\_{sup}$, no $\mathcal{L}\_{MMD}$, $\mathcal{L}\_{rec}$, $\mathcal{L}\_{sim}$),** there is no interaction between two sides of the Siamese architecture. *Our method performs similarly to the Base model.*
>     - Reflected on the loss curves: $\mathcal{L}\_{sim}=-0.17$ (domain factors are not similar, ideal value is -1.0); $\mathcal{L}\_{rec}=0$ (no reconstruction effect, ideal value is -2.0); $\mathcal{L}\_{MMD}=2.0$ (embedding spaces are not aligned, ideal value is 0).
> - **In case 2 (only $\mathcal{L}\_{sup}$,  $\mathcal{L}\_{MMD}$, no $\mathcal{L}\_{rec}$, $\mathcal{L}\_{sim}$):** There is no interaction between two sides of the Siamese architecture. *Our method performs similarly to the Base model.* *However, the MMD objective improves the orthogonal projection, which explains the slight improvements over **case 1**.*
>     - Reflected on the loss curves: $\mathcal{L}\_{sim}=-0.45$ (domain factors are somewhat similar, ideal value is -1.0); $\mathcal{L}\_{rec}=0$ (no reconstruction effect, ideal value is -2.0); $\mathcal{L}\_{MMD}=0$ (embedding spaces are aligned, ideal value is 0).
> - **In case 3 (only $\mathcal{L}\_{sup}$,  $\mathcal{L}\_{rec}$, no $\mathcal{L}\_{MMD}$, $\mathcal{L}\_{sim}$):** *The reconstruction objective ensures the interactions between two-sided pipeline, which is essential for learning the domain and domain-invariance representations.* We find that without $\mathcal{L}\_{MMD}$, our model can already learn to align the $\mathbf{v}$ and $\mathbf{z}$ embedding spaces and use orthogonal projection for prediction. According to the bar chart in Figure 9, *the $\mathcal{L}\_{rec}$ brings the most performance gains* compared to other objectives (exclude $\mathcal{L}\_{sup}$).
>     - Reflected on the loss curves: $\mathcal{L}\_{sim}=-0.45$ (domain factors are somewhat similar, ideal value is -1.0); $\mathcal{L}\_{rec}=-1.4$ (reconstruction effect, ideal value is -2.0); $\mathcal{L}\_{MMD}=0.15$ (embedding spaces are nearly aligned, ideal value is 0).
> - **In case 4 ($\mathcal{L}\_{sup}$, $\mathcal{L}\_{rec}$, $\mathcal{L}\_{MMD}$, no $\mathcal{L}\_{sim}$):** Adding $\mathcal{L}\_{MMD}$ can ensure the perfect alignment of the $\mathbf{v}$ and $\mathbf{z}$ embedding spaces and brings further improvements over **case 3**.
>     - Reflected on the loss curves: $\mathcal{L}\_{sim}=-0.45$ (domain factors are somewhat similar, ideal value is -1.0); $\mathcal{L}\_{rec}=-1.4$ (reconstruction effect, ideal value is -2.0); $\mathcal{L}\_{MMD}=0$ (embedding spaces are aligned, ideal value is 0).
> - Adding $\mathcal{L}\_{sim}$ for **case 4** leads to the final objective in our method.  $\mathcal{L}\_{sim}$ maximizes the similarity of the domain factors and brings further accuracy improvements.
> - In summary, $\mathcal{L}\_{sup}$ brings the label supervision (the most important one), $\mathcal{L}\_{rec}$ brings additional information (tells the model that two samples are from the same domain), $\mathcal{L}\_{MMD}$ improves the orthogonal projection (for better disentanglement), and $\mathcal{L}\_{sim}$ improves the learned domain factor (for better orthogonal projection).

---

> ### Author Response · Authors · 2022-11-17
> **Response to Reviewer TUM8 (Part II)**
>
> **Q2: In the “Remark for mutual reconstruction,” stating that z would not be incentivized to preserve label information requires a bit more support.**
>
> |                                          Cosine Similarity                                         | Seizure |  Sleep |  Drug  | Hospitalization |
> |:--------------------------------------------------------------------------------------------------:|:-------:|:------:|:------:|-----------------|
> | Weight($\mathbf{v}\rightarrow$labels) and Weight($\mathbf{v}\_{\perp\mathbf{z}}\rightarrow$ labels) | 0.8789  | 0.8251 | 0.5714 | 0.4627          |
> | Weight($\mathbf{v}\rightarrow$domains) and Weight($\mathbf{v}\_{//\mathbf{z}}\rightarrow$domains) | 0.2462  | 0.3273 | 0.1987 | 0.2111          |
> | Weight($\mathbf{v}\rightarrow$labels) and Weight($\mathbf{v}\rightarrow$domains)                   | 0.0182  | 0.0276 | 0.0445 | -0.0119         |
> | **Weight($\mathbf{v}\rightarrow$labels) vs Weight($\mathbf{v}\_{//\mathbf{z}}\rightarrow$labels)**    | **0.1151**  | **0.0773** | **0.0695** | **0.0524**          |
>
> Thanks. We have shown some relevant statistics of $\mathbf{z}$ in Section 4.4 and Appendix B.3 upon submission. During the rebuttal, we computed the cosine similarity of Weight($\mathbf{v}\rightarrow$labels) and Weight($\mathbf{v}\_{||\mathbf{z}}\rightarrow$labels) and added it as the fourth row in Table 3 (above). The low similarity value indicates that $\mathbf{v}\_{||\mathbf{z}}$ (or $\mathbf{z}$) contains little label information compared to $\mathbf{v}\_{\perp\mathbf{z}}$ (on the first row). It further verifies that our model learns domain representations and domain-invariant label representations separately in $\mathbf{v}\_{||\mathbf{z}}$ and $\mathbf{v}\_{\perp\mathbf{z}}$.
>
> ---
>
> We thank the reviewer again for your recognition and constructive comments. We are open to other insightful discussions.

---

### Official Review · Reviewer_Snq3 · 2022-10-26

**Confidence:** 3
**Correctness:** 3
**Technical Novelty And Significance:** 3
**Empirical Novelty And Significance:** 2
**Recommendation:** 5

**Clarity, Quality, Novelty And Reproducibility:**

Clarity: Most parts of the paper are written in a clear way.

Quality: The quality of the paper can be further improved.

Novelty: The novelty is fair.

Originality: Good.


**Strength And Weaknesses:**

S1: The methodology part is written in a clear way and well addressed the main claims in the introduction part.

S2: It is an interesting idea to treat each patient as a domain and address the more general many-domain generalization propblem.

S3: The extensive experiments on four healthcare datasets and two realistic and common
clinical settings demonstrates the effectiveness of the proposed model.

W1: It's concerned to involve the label information in steps 1 and 2, which causes higher prediction performance due to label information leakage.

W2: It is better to provide theoretical analysis, followed by Data Generative Model and Factorized Prediction Model.

W3: It is better to clarify mutual Reconstructions in section 3.3 about the features v′ hat and v hat.


**Summary Of The Paper:**

This paper addresses the many-domain generalization problem by treating each patient as a domain in healthcare applications. The paper proposes a model MANYDG to capture the domain and domain-invariant label representation by combining mutual reconstruction and orthogonal projection to explicitly remove the patient covariates. Experiments on four real-world datasets demonstrate the effectiveness of MANYDG.


**Summary Of The Review:**

Below the acceptance threshold.
Please see Strength And Weaknesses for detail.

---

> ### Author Response · Authors · 2022-11-17
> **Response to Reviewer Snq3**
>
> We thank the reviewer for the helpful feedback. We have uploaded a revision with the changes marked as blue. Our detailed responses are as follows:
>
> ---
>
> **Q1: It's concerned to involve the label information in steps 1 and 2, which causes higher prediction performance due to label information leakage.**
>
> This is a misunderstanding. STEP 1 applies a feature extractor from $\mathbf{x}$ to $\mathbf{v}$, and STEP 2 applies a domain encoder from $\mathbf{v}$ to $\mathbf{z}$. Both steps do not involve the label information. *The label information is involved only when calculating Objective 1, which happens in the training phase, but Objective 1 is not needed for the inference phase*. Thus, no label information is involved during inference.
>
> **Q2: It is better to provide theoretical analysis, followed by Data Generative Model and Factorized Prediction Model.**
>
> Thanks for the suggestion. This work provides extensive empirical evaluations of the proposed ManyDG method. In future works, we will consider theoretically analyzing our approach.
>
> **Q3: It is better to clarify mutual Reconstructions in section 3.3 about the features $\mathbf{v}′$ hat and $\mathbf{v}$ hat.**
>
> We rephrased this paragraph and updated it to the revision during the rebuttal. We can briefly recap the mutual reconstruction design here.
>
> - Our model trains on two samples from the same patient, denoted by $(\mathbf{x}, y)$, $(\mathbf{x}', y')$.
> - First, we obtain the feature vectors and domain factors by $\mathbf{h}\_{\theta}(\cdot)$ and $\mathbf{q}\_{\phi}(\cdot)$
>     - $(\mathbf{x}, y)$: feature vector $\mathbf{v}=\mathbf{h}\_{\theta}(\mathbf{x})$, domain factor $\mathbf{z}=\mathbf{q}\_{\phi}(\mathbf{v})$.
>     - $(\mathbf{x}', y')$: feature vector $\mathbf{v}'=\mathbf{h}\_{\theta}(\mathbf{x}')$, domain factor $\mathbf{z}'=\mathbf{q}\_{\phi}(\mathbf{v}')$.
> - Second, we use the decoder $\mathbf{p}_{\psi}(\cdot)$ for mutual reconstruction
>     - $\hat{\mathbf{v}'}=\mathbf{p}\_{\psi}(\mathbf{z},y')$ uses domain of $\mathbf{x}$ and label of $\mathbf{x}'$ for reconstructing $\mathbf{v}'$.
>     - $\hat{\mathbf{v}}=\mathbf{p}\_{\psi}(\mathbf{z}',y)$ uses domain of $\mathbf{x}'$ and label of $\mathbf{x}$ for reconstructing $\mathbf{v}$.
> - Then, we design the reconstruction objective $\mathcal{L}\_{rec}$. The “Remark” on Page 5 explains why reconstruction is a proper approach to ensure the learning of domain factor $\mathbf{z}$. We have also shown the empirical evidence in Section 4.4 and Appendix B.3.
>
> ---
>
> Thanks again for the valuable comments. Hope our rebuttal has addressed all your concerns.

---

### Official Review · Reviewer_e35g · 2022-10-31

**Confidence:** 3
**Correctness:** 3
**Technical Novelty And Significance:** 3
**Empirical Novelty And Significance:** 2
**Recommendation:** 8

**Clarity, Quality, Novelty And Reproducibility:**

The paper is nice to read and the method is simple which is a strength. The results seems promising with incremental improvements.

**Strength And Weaknesses:**

Strength:
- The paper is easy to read (except in some paragraphs)
- The method is simple

Comments:
The paper is nice to read, however, I have the following comments:

Sec 3.1: the method is applicable only if there is a label for each sample and this is not necessarily the case for healthcare application. In other words, there is one y^d for all samples x_i^d.

Sec 3.2: Motivation 2 was not lear at the beginning but after finishing step 2 it became clear. Maybe rewriting that part would be beneficial.

Sec 3.3:
  - what does this sentence mean "sg(·) means to stop gradient back-propagation for this quantity"? does it mean that you do not update v_mu and only update z?
  -  equation 11, should the denominator be norm(z)^2?

Experiments:
  - In table 3, is there any explanation why the second row has low similarity?
  - Results in Figure 3 and the message "our ManyDG consistently outperforms the baselines and shows increasing improvements as the number of labels decreases." are not convincing as the "rate" of deterioration is similar across all models.

**Summary Of The Paper:**

The authors propose a way to encode patient's data such that domain and personalized (or label) information are separated. This is achieved through domain generalization that assumes each patient as a domain. The method has multiple components, the first component to embed patients data into a different representation v. The second component extracts the domain information from v; this is learned through two objective functions: one for mutual reconstruction and the other for domain similarity. Then, by projecting v on z and removing the domain information from v, it can be used for predicting the label.



**Summary Of The Review:**

Read the strength and weakness section.

---

> ### Author Response · Authors · 2022-11-17
> **Response to Reviewer e35g**
>
> Thanks for your detailed suggestions. We have uploaded a revision and used blue to mark the new changes. Our responses are as follows.
>
> ---
>
> **Q1: the method is applicable only if there is a label for each sample, and this is not necessarily the case for healthcare applications. In other words, there is one $y^d$ for all samples $x_i^d$.**
>
> This is a good question. For cases where all patients only have one class, there will be no variations within each domain (the concept of “domains” will be meaningless). In such extreme cases, the DG models (including ours) may perform similarly to the Base ERM model. However, in many healthcare applications, the domains do exist, and they do not have to be patients. The domains can be geographical regions, hospitals, and doctors. And patients from each domain become samples. Our proposed method can still apply to those settings.
>
> **Q2: Motivation 2 was not clear at the beginning, but after finishing step 2, it became clear. Maybe rewriting that part would be beneficial.**
>
> Thanks for your suggestion. We have rephrased Motivation 2 and highlighted it in blue.
>
> **Q3: what does this sentence mean by "sg(·) means to stop gradient back-propagation for this quantity"? does it mean that you do not update $\mathbf{v}_\mu$ and only update $\mathbf{z}$?**
>
> sg(·) means `.detach()` in PyTorch. We only calculate the gradient for $||\mathbf{z}\_{\mu}- \mathbf{v}\_{\mu}||^2\_\mathcal{F}$ and treat the denominator as a float-based value (this part does not have gradient). Similar usages can be found in Figure 2 of [1] and Equation 3 of [2].
>
> [1] Grill, Jean-Bastien, et al. "Bootstrap your own latent-a new approach to self-supervised learning." NeurIPS 2020
>
> [2] Chen, Xinlei, and Kaiming He. "Exploring simple siamese representation learning." *CVPR* 2021.
>
> **Q4: In equation 11, should the denominator be the $\mbox{norm}(\mathbf{z})^2$?**
>
> Thanks very much for finding the typo! We updated it: $\mathbf{v}\_{||\mathbf{z}} = \mathbf{z} \cdot \langle\frac{\mathbf{v}}{||\mathbf{z}||},\frac{\mathbf{z}}{||\mathbf{z}||}\rangle$. Note that this equation is implemented in `supplementary/ManyDG/model.py#L961`, where we used the correct form when running the experiments.
>
> **Q5: In table 3, is there any explanation for why the second row has low similarity?**
>
>  We added the following explanations to the revision.
>
> > Compared to the first row (predicting the labels, e.g., 6-class classification in seizure detection), the second row corresponds to predicting one of the many domains (e.g., 2,702-class classification in seizure detection). Using the linear prediction model, the performances of the second row are naturally not as good as the first row. As a result, the learned coefficients in the second row are less similar than those learned in the first row. This may also explain why sleep staging has the highest similarity in the second row (since this task only has 78 domains).
>
> **Q6: Results in Figure 3 and the message "our ManyDG consistently outperforms the baselines and shows increasing improvements as the number of labels decreases." are not convincing as the "rate" of deterioration is similar across all models.**
>
> Thanks. The performance gap between our ManyDG and the best baseline does increase slightly with fewer training data, though it is not significant. And we have rephrased the result analysis in the revision according to your feedback.
>
>
> ---
>
> Thanks again for taking the time to review our paper and provide insightful feedback. We are open to discussion on other technical or experimental details.

---

### Official Review · Reviewer_FdB9 · 2022-11-01

**Confidence:** 4
**Correctness:** 3
**Technical Novelty And Significance:** 3
**Empirical Novelty And Significance:** 3
**Recommendation:** 8

**Clarity, Quality, Novelty And Reproducibility:**

As mentioned above, clarity is one of the strengths of this paper.
The proposed method seems novel enough and interesting.
The authors mentioned that the source codes are included in supplementary materials to which I don’t seem to have access. There’s detailed information on the experiment setup though.


**Details Of Ethics Concerns:**

Privacy and potential misuse of patient data are always risk factors in the healthcare domain. However, I would argue that the proposed method is not limited to this domain and does not pose more risk than similar use cases.

**Strength And Weaknesses:**

This paper is firstly very well structured, and the idea has been explained very clearly.

The introduced idea seems novel to me and I’m convinced of its application in such non i.i.d. data situations. I also appreciate the quantitative verification and the ablation study.

What I found less intuitive is the procedure of first learning v from x, z from v and then trying to decompose v into two directions parallel and orthogonal to z, respectively, since v is supposed to contain all information about z already. I’m wondering if one could simply decompose

v=z+u, v’=z+u’

where z encodes the domain information and u encodes the sample information - similar to the factor analysis.
Technically, we could use two objectives:

The similarity between encoder_z(x), encoder_z(x’) is to be minimized, and

The orthogonality between encoder_u(x) and encoder_u(x’) is to be maximized.


**Summary Of The Paper:**

This paper proposes to treat each patient as a domain and multiple observation of covariates as samples from the same domain. Consequently, one attempts to isolate the domain and sample specific features in the process of representation learning. Experiments on two datasets demonstrate competitive results compared with SOTA. This idea seems also generalizable beyond healthcare to other applications where such hierarchical assumption on the covariates may apply.

-----------------------------------------------------------------
Update: I very much appreciate the informative and detailed feedback from the authors.

**Summary Of The Review:**

In general this is a fairly good paper implementing assumptions on the data situation into the representation learning, which is highly generalizable beyond patient data.

---

> ### Author Response · Authors · 2022-11-17
> **Response to Reviewer FdB9 (Part I)**
>
> Thanks, we appreciate your recognition of our paper.
>
> This paper proposes the many-domain generalization setting and method. We use several healthcare applications to verify our proposals. Yes, the proposed setting and model are generalizable beyond the healthcare domain.
>
> The **.zip source codes** are included in supplementary materials upon submission. Please let us know if you still cannot access it, then we can re-upload the .zip file.
>
> ---
>
> ## **Performance comparision with the simple decomposition method**
>
> For creating dependencies between $\mathbf{z}$ and $\mathbf{u}$, the simple decomposition idea (proposed by the reviewer) needs to enforce the reconstruction of $\mathbf{x}$ by using $\mbox{encoder}\_\mathbf{z}(\mathbf{x})$ + $\mbox{encoder}\_\mathbf{u}(\mathbf{x})$.
>
> We follow the reviewer’s idea with the new reconstruction constraint and implement two simple decomposition baselines named SimD 1 and SimD 2. They have the same architecture (proposed by the reviewer and have different measures for the objectives). Details of these two baselines can be found in Appendix B.6. We conduct the experiments and show that our approach can have significant improvements over the simple decomposition baselines.
>
> |   Models   | Seizure (Accuracy) | Seizure (Kappa) | Seizure (Avg. F1) |
> |:----------:|:------------------:|:---------------:|:-----------------:|
> | SimD 1     | 0.5954 ± 0.0127    | 0.4792 ± 0.0197 | 0.4998 ± 0.0242   |
> | SimD 2     | 0.6499 ± 0.0153    | 0.5485 ± 0.0065 | 0.5686 ± 0.0092   |
> | **Our ManyDG** | **0.6754 ± 0.0079**    | **0.5627 ± 0.0066** | **0.6015 ± 0.0120**   |
>
> |     Models     | Sleep (Accuracy)    | Sleep (Kappa)       | Sleep (Avg. F1)     |
> |:--------------:|---------------------|---------------------|---------------------|
> | SimD 1         | 0.8862 ± 0.0015     | 0.7855 ± 0.0149     | 0.6962 ± 0.0025     |
> | SimD 2         | 0.9017 ± 0.0073     | 0.7978 ± 0.0205     | 0.6989 ± 0.0141     |
> | **Our ManyDG** | **0.9055 ± 0.0054** | **0.7998 ± 0.0124** | **0.7015 ± 0.0027** |
>
> |     Models     |    Drug (Jaccard)   |  Drug (Avg. AUPRC)  |    Drug (Avg. F1)   |
> |:--------------:|:-------------------:|:-------------------:|:-------------------:|
> | SimD 1         | 0.4894 ± 0.0087     | 0.7472 ± 0.0065     | 0.6482 ± 0.0192     |
> | SimD 2         | 0.4901 ± 0.0091     | 0.7536 ± 0.0115     | 0.6534 ± 0.0127     |
> | **Our ManyDG** | **0.5175 ± 0.0130** | **0.7746 ± 0.0035** | **0.6737 ± 0.0141** |
>
> |     Models     |  Hospitalization (AURPC) | Hospitalization (Kappa) | Hospitalization (Avg. F1) |
> |:--------------:|-------------------------|-------------------------|---------------------------|
> | SimD 1         | 0.6985 ± 0.0153         | 0.6368 ± 0.0026         | 0.5079 ± 0.0118           |
> | SimD 2         | 0.7047 ± 0.0105         | 0.6584 ± 0.0048         | 0.5260 ± 0.0103           |
> | **Our ManyDG** | **0.7258 ± 0.0132**     | **0.6752 ± 0.0025**     | **0.5531 ± 0.0144**       |
> ---
>
> ## **Why our architecture?**
>
> We elaborate on other drawbacks of the simple decomposition method and elaborate on how our ManyDG method solves/improves them. Assuming the information decomposition $\mathbf{v}=\mathbf{z}+\mathbf{u}$ (we use the reviewer’s notation), we can go over our considerations.
>
> ### Consideration 1: optimize $sim(\mathbf{z}, \mathbf{z'})$ is not enough
>
> To ensure that $\mathbf{z}$ and $\mathbf{z'}$ are the representations for the same domain, optimizing the similarity measure $sim(\mathbf{z}, \mathbf{z'})$ is like a **necessary condition** for learning $\mathbf{z}$. The model will likely learn to generate a constant vector for all $\mathbf{z}$ (regardless of the patients), which easily guarantees the best $sim(\mathbf{z}, \mathbf{z'})$. In our paper, we leverage that “two samples from the same domain share the domain information” and thus design the mutual reconstruction (like a  **sufficient condition**) for ensuring that $\mathbf{z}$ and $\mathbf{z'}$ are the domain representation (because they can be used to reconstruct a data from the domain).

---

> ### Author Response · Authors · 2022-11-17
> **Response to Reviewer FdB9 (Part II)**
>
> ### Consideration 2: first learn $\mathbf{v}$ from $\mathbf{x}$, then learn $\mathbf{z}$ from $\mathbf{v}$
>
> Some recent domain generalization or adaptation works [1][2][3][4] have demonstrated the success of sharing the first few layers between $\mbox{encoder}\_\mathbf{z}(\mathbf{x})$ and $\mbox{encoder}\_\mathbf{u}(\mathbf{x})$. This design can also reduce the number of trainable parameters. In our model, we use $\mathbf{h}\_{\theta}(\cdot)$ as the shared layers. Thus, $\mbox{encoder}\_\mathbf{z}(\mathbf{x})$ equals to $\mathbf{h}\_{\theta}(\cdot)$ plus $\mathbf{q}\_{\phi}(\cdot)$. For $\mbox{encoder}\_\mathbf{u}(\mathbf{x})$, we explain it in consideration 3.
>
> [1] Ganin et al. "Domain-adversarial training of neural networks." JMLR 2016
>
> [2] Zhao et al. "Learning sleep stages from radio signals: A conditional adversarial architecture." *ICML 2017*
>
> [3] Li et al. "Domain generalization with adversarial feature learning." *Proceedings of CVPR 2018*
>
> [4] Nam et al. "Reducing domain gap by reducing style bias." *Proceedings of the CVPR 2021*
>
> ### Consideration 3: how to decompose $\mathbf{v}$ into two directions
>
> Given the learned $\mathbf{z}$, our method aligns the embedding spaces of $\mathbf{v}$ and $\mathbf{z}$ (by MMD objective) and uses non-parametric orthogonal projection to obtain $\mathbf{u}$.
>
> **OPTION:** To obtain $\mathbf{u}$, we can also adopt the idea from the simple decomposition method, which needs to (i) add several layers after $\mathbf{h}\_{\theta}(\cdot)$ to learn $\mathbf{u}$ (i.e., $\mbox{encoder}\_\mathbf{u}(\mathbf{x})$ equals to $\mathbf{h}\_{\theta}(\cdot)$ plus these several layers) (ii) then, add soft orthogonality objective on $\mathbf{u}$ and $\mathbf{z}$; (iii) and then, add reconstruction objective on $\mathbf{v}$ by using $\mathbf{z}$ and $\mathbf{u}$.
>
> **Compared to our method, this OPTION needs more parameters and objective functions.** Performance-wise, our method works better than the simple decomposition methods, as we showed before.
>
> ## **Note on Ethics Flag**
>
> Thanks for bringing it to our attention. First, we have attached the License links and information for all datasets in Appendix A.4. The data usage in our study strictly follows the License regulations. Second, we have provided citations for all data sources and similar studies conducted on these datasets. Third, all the data used in this study are anonymous and de-identified. Last, as mentioned by the reviewer, the proposed method is not restricted to the healthcare domain.
>
> ---
>
> Thanks again for your time and appreciation. We are open to having more technical discussions on model architecture design.

---

### Official Review · Reviewer_xtPR · 2022-11-03

**Confidence:** 3
**Clarity, Quality, Novelty And Reproducibility:** Some parts of the paper are not justi…
**Correctness:** 2
**Technical Novelty And Significance:** 2
**Empirical Novelty And Significance:** 3
**Recommendation:** 3

**Strength And Weaknesses:**


Strength:

- There's an ablation study shows the contribution of each loss to the model performance.
- The paper experiments on several interesting datsets and applications.

Weakness:

- No definition of any of the variables. What dimensions they are?
- The mutual reconstruction seems unclear to me. "we take a new sample and its label (x',y') from the same domain as x".  Isn't the model "treating each patient as an individual domain"? Is the assumption of one patient having multiple samples mandetory?
- The motivation is unclear. The proposed predictor does not marginalize over z. Where does equation 2 come from? Also, why consider p(x|z,y) in motivation 2? It's irrelvant to motivation 1 and equation 2.
- It's unclear what the magnitudes of decomposed two orthogonal vectors are for each sample. It would be interesting to see a scatter plot for that.
- Suppose there is a strong spurious correlation between domain and labels. E.g. all the samples from some patients are all negative/positive, which actually happens a lot. In this case, the domain representation z should be able to predict the label very well. Will that cause a problem when you use the vector orthogonal to z?







**Summary Of The Paper:**

The paper assumes that each patient belongs to different domains given by some unobserved covariates. Mutual reconstruction is used to learn the domain covariates, which are then removed by orthogonal projection.  The proposed methods improve prediction performance on healthcare monitoring and EHR datasets.

**Summary Of The Review:**

The paper proposes an interesting method that treats each patient as a domain. However, different components of the model are not well explained.  E.g. what does z looks like? what does the model want to disentangle from the direction of z? Also, the writing is not rigours in maths.

---

> ### Author Response · Authors · 2022-11-17
> **Response to Reviewer xtPR (Part I)**
>
> We thank the reviewer for the constructive feedback. We have uploaded a revision and used blue to mark the new changes. Our detailed responses are as follows.
>
> ---
>
> **Q1: No definition of any of the variables. What dimensions they are?**
>
> Due to space limitation, we listed the detailed descriptions for every symbol in Appendix A.1. In the paper, variables $\mathbf{v}$ and $\mathbf{z}$ have the same dimensionality as mentioned under Equation 4. The other intermediate variables (such as $\mathbf{v}\_{||\mathbf{z}}$ and $\mathbf{v}\_{\perp\mathbf{z}}$) are the linear arithmetic results of $\mathbf{v}$ and $\mathbf{z}$, and thus they all have the same dimensionality as well. We are considering moving the symbol table to the main text and some experiments to the appendix.
>
> Furthermore, during the rebuttal, we added the following notation convention to Appendix A.1.
> > This paper uses the generic notation $\mathbf{x}$ to denote the input features. We use plain letters for scalars, such as $d$, $p$, $y$; boldface lowercase letters for vectors or vector-valued functions, e.g., $\mathbf{v}$, $\mathbf{z}$, $\mathbf{h}\_{\theta}(\cdot)$, $\mathbf{q}\_{\phi}(\cdot)$, $\mathbf{g}\_{\xi}(\cdot)$, $\mathbf{p}\_{\psi}(\cdot)$ and they all have the same dimensionality; Euler script letters for sets, e.g., $\mathcal{D}$, $\mathcal{S}$.
>
> **Q2: The mutual reconstruction seems unclear to me. "we take a new sample and its label (x', y') from the same domain as x.” Isn't the model "treating each patient as an individual domain"? Is the assumption of one patient having multiple samples mandatory?**
>
> Yes, we treat each patient as a domain. One domain (i.e., patient) can have multiple samples, as defined in Section 3.1 Problem Definition; No, the assumption “one patient having multiple samples” is not mandatory. In the case of domains (i.e., patients) with only one sample, we will skip the reconstruction loss $\mathcal{L}\_{rec}$ and the domain factor similarity loss $\mathcal{L}\_{sim}$. However, these samples still participate in calculating the supervised loss $\mathcal{L}\_{sup}$ and MMD loss $\mathcal{L}\_{MMD}$.
>
> **Q3: The motivation is unclear. The proposed predictor does not marginalize over z. Where does equation 2 come from?**
>
> Equation 2 is the definition of *marginal probability* conditioned on $\mathbf{x}$, which comes from
> $$
> \begin{equation}
> p(y|\mathbf{x}) =\int p(y,\mathbf{z}|\mathbf{x}) d\mathbf{z} = \int p(y|\mathbf{x}, \mathbf{z}) p(\mathbf{z}|\mathbf{x}) d\mathbf{z}.
> \end{equation}
> $$
>
> In real applications, people do not marginalize over $\mathbf{z}$. Instead, they use the approximation
> $$
> \begin{equation}
> p(y|\mathbf{x})= \int p(y|\mathbf{x}, \mathbf{z}) p(\mathbf{z}|\mathbf{x}) d\mathbf{z}\approx\frac{1}{N}\sum\_{i=1}^N p(y|\mathbf{x}, \mathbf{z}_i),
> \end{equation}
> $$
> sampling $\mathbf{z}$ multiple times from $p(\mathbf{z}|\mathbf{x})$ and using the average effect (1/N) of these $\mathbf{z}$ for prediction (in our case and [2]) or generation (in VAE [1][3]). The same techniques are widely used, such as Equation 22 in [1], Section 2.1 in [2], and Equation 5 in [3]. One typically uses N=1 in VAE [1], and we also use N=1 in our approach: generate one $\mathbf{z}$ from $p(\mathbf{z}|\mathbf{x})$ and then use $p(y|\mathbf{x},\mathbf{z})$ for prediction.
>
> [1] Kingma, Diederik P., and Max Welling. "Auto-encoding variational bayes." *arXiv preprint arXiv:1312.6114* (2013).
>
> [2] Wiles, Olivia, et al. "A fine-grained analysis on distribution shift." ICLR 2022.
>
> [3] Higgins, Irina, et al. "beta-vae: Learning basic visual concepts with a constrained variational framework." ICLR 2017.

---

> > ### Comment · Reviewer_xtPR · 2022-11-18
> > **Feedback**
> >
> > I thank the authors for the explanation and extra experiments. These concerns still hold.
> >
> > 1. The maths is incorrect. In the paper, z is deterministic given x, because $z=q_{\phi}(h_{\theta}(x))$ and q, h are all deterministic functions. Then in Equation 2, how is p(z|x) defined? This is so different from VAE paper, where z was defined as a random variable so the integral can be approximated by sampling. However, since here z is fixed given x, I don't see the point of p(z|x) and how to sample z_i.
> >
> > Also, VAE proves that optimizing the loss function is optimizing the lower bound of the marginal likelihood. However, the proposed loss functions are irrelevant to the probabilistic model.
> >
> > 2. The interpretation of z and v// is still not clear.
> > In Table 3, Weight(v to domains) and Weight(v// to domains) is not highly correlated. It shows that v// and z cannot represent the domain quite well. This does not support the claim.

---

> > > ### Author Response · Authors · 2022-11-18
> > > **Thanks for the response. Let us quickly clarify these minor confusions**
> > >
> > > **For Question 1:**
> > >
> > > First, our method can be probabilistic or deterministic. Performance-wise, we have tried the probabilistic form before. We found that its performance is unstable and is consistently worse than our deterministic method. Also, our deterministic method is simple and more efficient.
> > >
> > > Second, the probabilistic integral in Equation 2 is just mentioned for illustration purposes, which inspires us to learn domain $\mathbf{z}$ for the prediction task. The math of the motivation is correct, but we used a deterministic implementation based on the prediction setting (although motivated by the probabilistic form in the first place).
> > >
> > > Third, our method differs from VAE in that we target the prediction task. VAE is commonly used for the generation task, and it deterministically learns the $\mu$ and $\sigma$ vectors from $\mathbf{x}$ and sample a $\mathbf{z}$ from $\mbox{Normal}(\mu, \sigma)$ by reparameterization. In our current architecture, we intentionally remove $\sigma$ (the randomness part) and let $\mu$ be the domain vector $\mathbf{z}$. Thus, we have different objectives instead of ELBO.
> > >
> > > We also add clarification in Section 3.3 (revision) that our model is different and we do not need sampling.
> > >
> > > **For Question 2:**
> > >
> > > In Table 3, the second row is relatively lower than the first row, but it does not mean “v// and z cannot represent the domain quite well.” Compared to the first row (predicting the labels, e.g., 6-class classification in seizure detection), the second row corresponds to predicting one of the many domains (e.g., 2,702-class classification in seizure detection). As a result, the learned coefficients in the second row are naturally less similar than in the first row. This may also explain why sleep staging task has the highest similarity in the second row (since this task only has 78 domains).
> > >
> > > In the case of large label space (e.g., 2,702-class), our “cosine similarity” proxy may not fully reflect the correlation. We have also shown that the within-domain $\mathbf{z}$ similarity is higher than the cross-domain $\mathbf{z}$ similarity in Table 9, which can support our claim as well. We are willing to show other proxy statistics if there is any suggestion.
> > >
> > > ---
> > >
> > > Thanks again, we hope our explanations can address your concerns.

---

> ### Author Response · Authors · 2022-11-17
> **Response to Reviewer xtPR (Part II)**
>
> **Q4: Also, why consider $p(\mathbf{x}|\mathbf{z},y)$ in motivation 2? It's irrelevant to motivation 1 and equation 2.**
>
> Motivation 2 is related to motivation 1 and Equation 2. As mentioned in **Q3**, the marginalization step over $\mathbf{z}$ is commonly replaced by learning a $\mathbf{z}$ from $p(\mathbf{z}|\mathbf{x})$. Thus, we need Motivation 2 (mutual reconstruction) to guide the learning process of $\mathbf{z}$ for better prediction in Motivation 1. Our Motivation 2 (reconstructing another sample from the same domain) is inspired by the reconstruction design in VAE [1] (reconstructing the same sample).
>
> During the rebuttal, we have also rephrased Motivation 2 to improve clarity.
>
> **Q5: It's unclear what the magnitudes of decomposed two orthogonal vectors are for each sample. It would be interesting to see a scatter plot for that.**
>
> Following your suggestion, we added the scatter plots to Appendix B.5.
>
> **Q6: Suppose there is a strong spurious correlation between domain and labels. E.g., all the samples from some patients are all negative/positive, which happens a lot. In this case, the domain representation z should be able to predict the label very well. Will that cause a problem when you use the vector orthogonal to z?**
>
> Thanks for bringing up this point. *This does not cause a problem for our method
>  in practice*.
>
> First, our method is proposed to solve such problems. For example, some patients in the IIIC seizure data have only one seizure type (similar to the example in the question). Some patients in the SLEEP-EDF may have insomnia diseases and are of different ages, and thus their sleep cycles will be different. These are the real spurious correlations in our healthcare datasets. Extensive experiments verify that our proposed model outperforms the baselines in such scenarios.
>
> Second, we demonstrate whether the domain representation $\mathbf{z}$ can predict the labels well and add the fourth row to Table 3 (shown below), which reports the cosine similarity between Weight($\mathbf{v}\rightarrow$labels) and Weight($\mathbf{v}\_{||\mathbf{z}}\rightarrow$labels). The results show that $\mathbf{v}\_{||\mathbf{z}}$ (or $\mathbf{z}$) may contain partial label information (the values are all slightly above 0.05) due to some “strong spurious correlation” samples. However, they cannot predict the labels very well in either healthcare monitoring data or EHR datasets.
>
> |                                          Cosine Similarity                                         | Seizure |  Sleep |  Drug  | Hospitalization |
> |:--------------------------------------------------------------------------------------------------:|:-------:|:------:|:------:|-----------------|
> | Weight($\mathbf{v}\rightarrow$labels) and Weight($\mathbf{v}\_{\perp\mathbf{z}}\rightarrow$ labels) | 0.8789  | 0.8251 | 0.5714 | 0.4627          |
> | Weight($\mathbf{v}\rightarrow$domains) and Weight($\mathbf{v}\_{//\mathbf{z}}\rightarrow$domains) | 0.2462  | 0.3273 | 0.1987 | 0.2111          |
> | Weight($\mathbf{v}\rightarrow$labels) and Weight($\mathbf{v}\rightarrow$domains)                   | 0.0182  | 0.0276 | 0.0445 | -0.0119         |
> | **Weight($\mathbf{v}\rightarrow$labels) and Weight($\mathbf{v}\_{//\mathbf{z}}\rightarrow$labels)**   | **0.1151**  | **0.0773** | **0.0695** | **0.0524**          |
>
>
> Third, if all samples from the same patient have the same labels, then there are no variations within each domain (the concept of “domain” is meaningless in this case). For these extreme cases, all domain generalization models may not be applicable.
>
> **Q7: what does z look like? what does the model want to disentangle from the direction of z?**
>
> $\mathbf{z}$ is a learned domain vector that contains the target-irrelevant domain information. The model wants to remove this information and obtain cleaner feature vectors $\mathbf{v}_{\perp\mathbf{z}}$ for prediction. To provide more insights, we added scatter plots of $\mathbf{z}$ in Appendix B.5.
>
> ---
>
> Thanks again for your time and valuable questions. We hope our new results and explanations can clear your concerns. We are happy to explain any other component of the model.

---

### Decision · Program_Chairs · 2023-01-20

**Decision:**

Accept: poster

**Justification For Why Not Higher Score:**

There is just enough support for accept.

**Justification For Why Not Lower Score:**

There is just enough support for accept.

**Metareview: Summary, Strengths And Weaknesses:**

This paper proposes a new domain generalization method, termed ManyDG, for healthcare applications. Unlike previous method, ManyDG treats each patient is as a separate domain. A Siamese architecture is used to learn a latent representation (v) and a domain representation (z) for each patient using a mutual reconstruction loss on v from pairs of training examples of the same patient. Domain-dependent information in v is removed via orthogonal project, and what remains is used for prediction.  The reviewers find the paper easy to read, the work interesting, and the empirical evaluations compelling, although they also think that some ablation studies are missing.  Overall, there is sufficient support to accept the paper.

**Note From Pc:**

if the above contains the word "oral" or "spotlight" please see: "oral" presentation means -> notable-top-5% and "spotlight" means -> notable-top-25%. As stated in our emails, we are disassociating presentation type from AC recommendations